# Impact of ADCP motion on structure function estimates of turbulent kinetic energy dissipation rate

Brian D. Scannell[1], Yueng-Djern Lenn[1], and Tom P. Rippeth[1]

[1]School of Ocean Sciences, Bangor University, Menai Bridge, Ynys Môn, LL59 5AB, United Kingdom

**Correspondence:** Brian D. Scannell (brian.scannell@bangor.ac.uk)

**Abstract.** Turbulent mixing is a key process in the transport of heat, salt and nutrients in the marine environment, with fluxes commonly derived directly from estimates of the turbulent kinetic energy dissipation rate, $\varepsilon$. Time series of $\varepsilon$ estimates are therefore useful in helping to identify and quantify key biogeochemical processes. The velocity structure function method can be used to determine time series of $\varepsilon$ estimates using along-beam velocity measurements from suitably configured acoustic Doppler current profilers (ADCP). Shear in the background current can bias such estimates, therefore standard practice is to deduct the mean or linear trend from the along-beam velocity over the period of an observation burst. This procedure is effective if the orientation of the ADCP to the current remains constant over the burst period. However, if the orientation of the ADCP varies, a proportion of the velocity difference between bins is retained in the structure function and the resulting $\varepsilon$ estimates will be biased. Long-term observations from a mooring with three inline ADCP show the heading oscillating with an angular range that depends on the flow speed; from large, slow oscillations at low flow speeds to smaller, higher frequency oscillations at higher flow speeds. The mean tilt was also determined by the flow speed, whilst the tilt oscillation range was primarily determined by surface wave height. Synthesised along-beam velocity data for an ADCP subject to sinusoidal oscillation in a sheared flow indicates that the retained proportion of the potential bias is primarily determined by the angular range of the oscillation, with the impact varying between beams depending on the mean heading relative to the flow. Since the heading is typically unconstrained in a tethered mooring, heading oscillation is likely to be the most significant influence on the retained bias for a given level of shear. Use of an instrument housing designed to reduce oscillation would mitigate the impact, whilst if the shear is linear over the observation depth range, the bias can be corrected using a modified structure function method designed to correct for bias due to surface waves.

## 1 Introduction

The most well established technique for making observations of the turbulent kinetic energy (TKE) dissipation rate, $\varepsilon$, uses microstructure profilers (e.g. Lueck, 2016). The approach produces high resolution vertical profiles of $\varepsilon$, but is expensive as it requires a surface vessel and staff, as well as being limited in the sampling interval and duration of observations and the conditions under which they can be made. An alternative approach using Acoustic Doppler Velocimeters (ADV) to make point observations of the velocity spectrum has been used from a mooring, but $\varepsilon$ estimates are subject to potentially high levels of motion induced contamination (Bluteau et al., 2016). In comparison, the velocity structure function method offers the potential

of generating time series of $\varepsilon$ estimates using industry standard acoustic Doppler current profiler (ADCP) instruments, which are relatively cheap, robust and designed for long-term deployment under the widest range of environmental conditions.

Standard ADCP have three or four beams, each oriented at a common beam angle to the instrument axis. If the instrument is vertical, the velocity field can then be determined (Teledyne RD Instruments, 2010). Recent developments have seen the introduction of ADCP with an additional beam oriented along the instrument axis, allowing direct observation of the vertical velocity (e.g. Togneri et al., 2017; Guerra and Thomson, 2017).

The structure function method for estimating $\varepsilon$, (Wiles et al., 2006), derives from the Kolmogorov hypotheses of similarity and local isotropy in high Reynolds number flows (Kolmogorov, 1991a, b, translated from the original 1941 Russian publications). Originally used for observations of atmospheric turbulence, the technique is now established as a means of acquiring long-term observations of $\varepsilon$ in the aquatic environment under a wide range of conditions (e.g. Lucas et al., 2014; McMillan and Hay, 2017; Buckingham et al., 2019; Simpson et al., 2021).

The method determines $\varepsilon$ as a function of the difference in the along-axis turbulent velocity with the spatial separation of the observation points. This is readily applied to ADCP, which by design measure the radial (along-beam) velocity at defined separation distances. The detection limit and resolution are inherently determined by the uncertainty of the velocity measurements, which depend on manufacturers' proprietary techniques and are not published in a consistent form. However, the development of new ADCP operating modes such as pulse-pulse coherent and high ping rates has allowed high spatial resolution, low variance velocity measurements to be made without the need for extensive time averaging, but with limited beam range. This has encouraged innovations such as deployments on tethered moorings to acquire turbulence measurements in sections of the water column important for mixing (e.g. Lucas et al., 2014; Simpson et al., 2015; Buckingham et al., 2019), and on surface drifters to provide quasi-synoptic observations of the spatial distribution of turbulence (e.g. Guerra and Thomson, 2017; Guerra et al., 2021).

Standard practice is to assume that any non-turbulent velocity differences between bins are static or slowly varying, such that they can be excluded by deducting the the mean or linear trend over a burst of profiles for each bin (Wiles et al., 2006; McMillan and Hay, 2017). It is then assumed that all residual velocity differences are turbulent.

Shear in the background flow is a potential source of non-turbulent velocity differences between bins for standard ADCP angled beams. If the ADCP is on a static bedframe, the orientation of the beams to the background flow will be constant over the burst period and the standard procedure will fully remove the non-turbulent velocity difference between bins due to the sheared flow. Similarly, for a static vertical beams, the along-beam velocity is independent by any shear in the background flow, therefore no velocity difference between bins arises.

However, an ADCP on a tethered mooring is typically free to rotate about its vertical axis, so that the heading varies. Drag on the mooring and the instrument may also result in the instrument tilt varying, resulting in differences in the vertical range and the orientation of the beams, whilst surface waves may affect the instruments directly or by varying the tension and shape of the mooring. Similarly, ADCP deployed on surface drifters are free to rotate about their vertical axis, whilst surface waves may cause periodic variation in the instrument tilt.

Velocities due to the rotation of a tethered or drifter-mounted ADCP are normal to the beams (both angled and vertical) and therefore do not directly contribute to the observed along-beam velocities (Lucas et al., 2014; Zippel et al., 2020). However changes in the ADCP orientation will result in a variation in the background flow contribution to the along-beam velocity, with angled beams affected by changes in both heading and tilt, whilst vertical beams will only be affected by changes in tilt. The magnitude of the background flow contribution to the along-beam velocity increasing as the beam becomes more

closely aligned with the flow and vice versa. The burst mean will therefore underestimate the contribution for those profiles when the beam is most closely aligned with the background flow and overestimate it at other times. Deducting the burst mean cannot fully remove this time-varying contribution. If the flow is sheared, a proportion of the associated non-turbulent velocity difference between bins is unavoidably retained, contributing to the structure function and biasing the resulting $\varepsilon$ estimates.

This is similar to the effect of the vertical gradient of the orbital velocity forced by surface gravity waves, which can lead

to non-turbulent velocity differences between bins being retained in the structure function and potential bias in $\varepsilon$ estimates (Whipple and Luettich, 2009; Scannell et al., 2017).

The aims of this paper are, firstly, to demonstrate that $\varepsilon$ estimates derived from velocity observations from the angled beams of a tethered ADCP in a sheared flow using the standard structure function method are inherently susceptible to bias if the instrument orientation to the flow varies; to highlight the key factors determining the level of such bias; and to outline possible

means of mitigating or correcting for the effect. The principles equally apply to the vertical beam of ADCP subject to tilt such that the background flow contributes a periodic component to the along-beam velocity. Whilst the specific impact has not been evaluated, the same conclusions apply. Section 2 briefly outlines the structure function methodology and considers the scaling of the potential $\varepsilon$ bias arising due to linear shear in the background flow. Section 2 describes observations from a mooring in the central Celtic Sea with three tethered ADCP at different depths to illustrate how the motion of the ADCP varies with both

the flow speed and the amplitude of surface waves. Section 4 uses synthetic data to examine the dependence on the level of retained $\varepsilon$ bias on the ADCP motion. Finally, section 5 is a discussion of the findings and the potential for correcting the bias.

## 2 Potential Bias

### 2.1 Structure Function Method

Figure 1 illustrates the geometry for a Teledyne RDI WorkHorse four-beam ADCP, which is similar to that for other instru-

85 ments. Based on a standard Cartesian coordinate framework $(x, y, z)$ relative to the transducer head, each of the beams is tilted at beam angle $\theta$ to the along-instrument $z$-axis, with beams 1 and 2 oriented either side of the $z$-axis in the $x = 0$ plane and beams 3 and 4 similarly positioned in the $y = 0$ plane. Instrument orientation and motion can then be described in terms of heading, $\phi_H$; pitch, $\phi_P$; and roll, $\phi_R$, being the rotation angles about the $z$-, $x$- and $y$-axes respectively. Along-beam velocities, $b$, (positive towards the transducer) are measured for volume bins centred at fixed distances (time range gates) from the

90 transducer, such that the $z$ coordinate is the same for bin $n$ in each beam. The $z$-axis separation distance between bin centres, $\delta z$, is the same for all beams and bins, with the along-beam separation distance between adjacent bins in any beam being $\delta r = \delta z / \cos \theta$.

By observing the along-beam velocities at fixed separation distances, ADCP provide the information required for independent longitudinal structure function calculations for each beam. The theoretical basis of the method is described in detail elsewhere (e.g. Pope, 2000). Applied to a burst of ADCP observations comprising $N$ sets of along-beam velocity profiles, $b(i,j,k)$, where $i$ is the beam number, $j$ is the bin number and $k$ is the profile number ($1 \leqslant k \leqslant N$), the turbulent velocity, $b'$, is typically calculated as:

$$b'(i,j,k) = b(i,j,k) - \langle b(i,j) \rangle \tag{1}$$

the angle brackets indicating the mean of $b(i,j)$ over the $N$ profiles in the burst (Wiles et al., 2006). An alternative approach is to deduct the linear trend of $b$ over the burst, allowing for a steady variation in the speed of the background flow (McMillan and Hay, 2017).

The second-order structure function, $D_{LL}$, for along-beam separation distance $r_n = n\delta r$, where $n$ is the number of bins separating the observations, is then evaluated using a bin-centred difference scheme as:

$$D_{LL}(i,j,r_n) = \left\langle \left[ b'\left(i,j-\tfrac{n}{2}\right) - b'\left(i,j+\tfrac{n}{2}\right) \right]^2 \right\rangle \tag{2}$$

the angle brackets again indicating the arithmetic mean across the $N$ profiles in the burst (Wiles et al., 2006). For odd $n$, the mean of the two offset bin difference options is taken. This approach yields individual $D_{LL}(i,j,r_n)$ values, allowing a vertical profile of $\varepsilon$ estimates to be constructed (e.g. Simpson et al., 2015). An alternative approach evaluates all possible $r_n$ for a range of bins to give a representative value for the depth range (McMillan and Hay, 2017).

The Kolmogorov hypotheses anticipate that $D_{LL}(r)$ should vary solely as a function of $\varepsilon$ and $r$ as:

$$D_{LL}(r) = C_2\, \varepsilon^{\frac{2}{3}}\, r^{\frac{2}{3}} \tag{3}$$

with $C_2$ being an empirical constant, for which atmospheric studies suggest a value of $2.1 \pm 0.1$ (Sauvageot, 1992), whilst laboratory measurements of grid turbulence in high Reynolds number flows give a value of $2.0 \pm 15\%$ (Sreenivasan, 1995). The appropriate value is also potentially influenced by Reynolds number, anisotropy of the turbulent eddies and proximity to a boundary (Jabbari et al., 2016). Studies commonly adopt values of 2.1 (e.g. Lorke, 2007; Lucas et al., 2014; Wiles et al., 2006) or 2.0 (e.g. McMillan and Hay, 2017; Simpson et al., 2021).

Doppler noise associated with the velocity observations introduces an offset, hence standard practice is to use a least-squares linear regression of $D_{LL}$ against $r^{\frac{2}{3}}$ as:

$$D_{LL}(r) = a_0 + a_1 r^{\frac{2}{3}} \tag{4}$$

the intercept $a_0$ typically being taken as twice the Doppler noise variance of the velocity measurements, although McMillan and Hay (2017) demonstrate a dependence on $\varepsilon$ levels, with $a_0$ decreasing with increasing $\varepsilon$.

The gradient $a_1$ from equation (4) is then used to determine $\varepsilon$ as:

$$\varepsilon = \left( \frac{a_1}{C_2} \right)^{\frac{3}{2}} \tag{5}$$

The linear regression is evaluated for $r_n \leqslant r_{\max}$, which is required to be less than the spatial scale over which the isotropic turbulence assumption in the Kolmogorov hypotheses is considered to be valid. In practice there may be a trade-off between limiting the spatial scale and increasing the number of data points to improve confidence in the linear regression.

Scannell et al. (2017) describe a modified method to correct for the bias due to the spatial gradient of the orbital velocities associated with surface gravity waves. The periodic nature of the wave-forced contribution to the along-beam velocity, $\widetilde{b}$, means that it is wholly retained in $b'$. Over a limited spatial scale, the velocity difference between bins, $\delta\widetilde{b}(r)$, varies approximately linearly with $r$, hence the contribution to $D_{LL}$ varies as $r^2$. Modifying the regression equation (4) with the inclusion of an additional term as:

$$D_{LL}(r) = a_0 + a_1\, r^{\frac{2}{3}} + a_3\, (r^{\frac{2}{3}})^3 \tag{6}$$

allows the turbulent contribution, described by $a_1$, to be isolated from the non-turbulent component due to the wave orbital velocity.

## 2.2 Potential Impact of Shear

For an upward- or downward-looking ADCP with constant heading such that the horizontal projection of beam $i$ is oriented into a steady, non-turbulent, vertically-sheared horizontal flow with current speed $U(z)$, the difference in the along-beam velocity $b$ observed between bin number $j$ and $j + n$ will be:

$$b(i,j) - b(i,j+n) = \sin\theta\, n\, \delta z\, \frac{\partial U}{\partial z} \tag{7}$$

where $\theta$ is the ADCP beam angle (from the instrument axis) and $\delta z$ is the vertical bin centre separation distance of the velocity measurement bins. Calculating the structure function with $b$ rather than $b'$ fully retains these non-turbulent velocity differences, such that:

$$D_{LL}^b(i,j,r_n) = r_n^2 \sin^2\theta \cos^2\theta \left(\frac{\partial U}{\partial z}\right)^2 \tag{8}$$

The standard method linear regression of $D_{LL}^b$ against $r^{\frac{2}{3}}$ as per equation (4) yields gradient $a_1$, with equation (5) giving the potential bias TKE dissipation rate, $\varepsilon^b$.

Figure 2 illustrates the variation of $\varepsilon^b$ for an ADCP with a $20°$ beam angle (standard for the Teledyne RDI WorkHorse), with the vertical bin size $\delta z$ varying between $0.1\,\mathrm{m}$ and $0.5\,\mathrm{m}$; the maximum separation distance used in the regression, $r_{\max}$, varying between 5 and 25 bins; and shear-squared, $S^2 = \left(\frac{\partial U}{\partial z}\right)^2$, of $1 \times 10^{-5}\,\mathrm{s}^{-2}$ and $1 \times 10^{-4}\,\mathrm{s}^{-2}$. For each permutation, $\varepsilon^b$ is calculated for a beam directly aligned with the sheared flow and for those bins for which all $r \leqslant r_{\max}$ are evaluated. Such bin sizes and $r_{\max}$ configurations are consistent with deployments in regions where mixing is of interest, such as the pycnocline in shelf seas, where shear levels frequently exceed $1 \times 10^{-4}\,\mathrm{s}^{-2}$ and $\varepsilon$ levels are commonly in the range $1 \times 10^{-9}\,\mathrm{W\,kg^{-1}}$ to $1 \times 10^{-7}\,\mathrm{W\,kg^{-1}}$ (e.g. Palmer et al., 2013).

Since $D_{LL}^b$ exhibits a linear dependence on $S^2$, $a_1$ also varies linearly with $S^2$, hence $\varepsilon^b$ varies as $S^3$. Consequently, increasing $S^2$ from $1 \times 10^{-5}\,\mathrm{m^2\,s^{-2}}$ to $1 \times 10^{-4}\,\mathrm{m^2\,s^{-2}}$ increases $\varepsilon^b$ by a factor of $10^{\frac{3}{2}}$ for all $r_{\max}$ and $\delta z$ options.

The $r^2$ length-scale dependency of $D_{LL}^b$ means that the standard method regression of equation (4) is imposing a least-squares linear fit against $r^{\frac{2}{3}}$ to a term varying as $(r^{\frac{2}{3}})^3$. The gradient $a_1$ and hence $\varepsilon^b$ therefore increases rapidly with $r_{\max}$, whilst reducing the bin size increases the number of evaluated distances for a given $r_{\max}$, slightly reducing $a_1$ and $\varepsilon^b$.

Whilst $\varepsilon^b$ can be derived for a known instrument configuration and anticipated shear, it is a theoretical maximum bias affecting beams directly aligned with the sheared flow and assuming all of the shear-related non-turbulent velocity difference between bins propagates through to the calculated structure function. The actual bias in the resolved $\varepsilon$ values will be a fraction of $\varepsilon^b$ determined by the proportion of the non-turbulent velocity differences between bins due to the shear retained in $b'$ as a consequence of the motion of the ADCP. Section 3 therefore uses long-term data on moored ADCP configured for turbulence observations to examine how the motion of a tethered ADCP is influenced by the environmental conditions.

Quantifying the retained proportion of $\varepsilon^b$ under a wide range of ADCP motion scenarios when using the standard regression method, together with testing the effectiveness of the modified regression method based on equation (6) at reducing the bias, is then evaluated using synthesised velocity data in section 4.

## 3 Field Observations of ADCP Motion

This section examines the heading and tilt sensor data from three in-line tethered ADCP deployed on a buoyancy-tensioned mooring at a site in the Celtic Sea with a water depth of $145\,\text{m}$ over a sixteen month period, providing data under a wide range of current and wave conditions. Details of the deployments and the data return together with information on the heading and tilt observations are given in Appendix A.

### 3.1 Moorings

Three Teledyne RD Instruments (TRDI) $600\,\text{kHz}$ Workhorse ADCP were deployed, with the nominal depths of the upper, middle and lower instruments being $20\,\text{m}$, $33\,\text{m}$ and $50\,\text{m}$ respectively. The upper and lower instruments were deployed upwards-looking in spherical syntactic buoys, whilst the middle instrument was deployed downward-looking in an open frame as illustrated in Figure 3. All had four-beam, Janus-style transducer heads, with the upper and middle instruments having a $20°$ beam angle and the lower a $30°$ beam angle. The same configuration was used for all instruments and deployment periods, with a vertical bin size of $10\,\text{cm}$ and the first bin centred $0.97\,\text{m}$ vertically from the transducer head. Pulse-pulse coherent (TRDI mode 5) single-ping ensemble (no averaging) observations of along-beam velocity were made at $1\,\text{Hz}$ for $5\,\text{min}$ followed by $15\,\text{min}$ sleep, yielding 3 bursts of observations per hour, each comprising 300 profiles for each beam. Velocities were typically resolved for bins 1 to 32 (1 to 29) for the $20°$ ($30°$) beam angle instruments, consistent with the expected range for the operating mode (Teledyne RD Instruments, 1999).

Three-axis orientation data was recorded for each profile, providing a description of the instrument motion during each observation burst. As illustrated in Figure 1, heading, $\phi_H$ (°N), is the rotation about the vertical axis expressed as the compass direction of the horizontal projection of beam 3, whilst tilt sensors describe the rotation about the horizontal axes, with pitch,

$\phi_P$ (°), being rotation in the plane of beams 3 and 4; and roll, $\phi_R$ (°), being rotation in the plane of beams 1 and 2, with both $\phi_P$ and $\phi_R$ being zero indicating that the instrument is vertical (Teledyne RD Instruments, 2010).

The along-beam velocity data for each profile was converted to earth coordinates following Teledyne RD Instruments (2010). The burst mean horizontal velocities were depth-averaged over the $\sim 3\,\mathrm{m}$ range of the observations and the dominant tidal constituents identified using the U-Tide Matlab functions (Codiga, 2011). The site is characterised by clockwise rotating semi-diurnal tides, with a pronounced spring-neap variation. Over the full deployment period, the horizontal current speed, $U$, observed by the upper instrument had a median value of $0.28\,\mathrm{m\,s^{-1}}$, with $U \leqslant 0.1\,\mathrm{m\,s^{-1}}$ for just $4.1\,\%$ of observations. The implication being that the ADCP mooring was under almost continual drag, rotating semi-diurnally about the position of the anchor weight.

A UK Met Office Ocean Data Acquisition System (ODAS) buoy, together with a Triaxys frequency-direction wave buoy, were moored less than $1\,\mathrm{km}$ away, providing hourly meteorological data, wave statistics and spectra. Significant wave height was derived from the wave spectra data as:

$$H_{m_0} = 4 \sqrt{\sum_{n=1}^{32} S_f(n)\, \delta f(n)} \tag{9}$$

where $n$ is the wave frequency band number; $S_f(n)$ is the surface displacement variance (or "wave energy density") per unit frequency $(\mathrm{m^2\,Hz^{-1}})$ for band $n$; and $\delta f(n)$ is the width of the frequency band (Hz). The 32 frequency bands of the Triaxys buoy having central frequencies between $0.03\,\mathrm{Hz}$ and $0.6\,\mathrm{Hz}$ with widths increasing from $0.005\,\mathrm{Hz}$ to $0.08\,\mathrm{Hz}$.

The annual median $H_{m_0}$ was $2.54\,\mathrm{m}$, with $90\,\%$ of observations $\geqslant 1.25\,\mathrm{m}$ and $10\,\% \geqslant 5.03\,\mathrm{m}$. There was a significant seasonal variation, with over $90\,\%$ of observations during the "summer" deployment 2 (19[th] June to 21[st] August 2014) being less than the annual median, and almost $23\,\%$ of observations during the "winter" deployment 4 (21[st] November 2014 to 4[th] April 2015) exceeding the annual 90[th] percentile.

## 3.2 ADCP Motion Sample

Panels (a) and (b) of Figure 4 show sample data for a $30\,\mathrm{hour}$ period, with the solid lines in panel (a) showing the depth-averaged burst mean horizontal current speed, $U$, and the markers showing the compass direction (to), $\Phi$, from the earth coordinate velocity calculated for each burst profile, with the colour indicating the instrument. Panel (b) shows the $\phi_H$ data for each instrument for all bursts over the same period.

All three instruments are in close agreement for $U$, which varies over the range $0.2\,\mathrm{m\,s^{-1}}$ to $0.5\,\mathrm{m\,s^{-1}}$. Current direction, $\Phi$, shows the tide rotating clockwise, with the $U$ maxima coinciding with the flow being towards the South-West and the North-East. For the upper and lower instruments, $\Phi$, is in good agreement throughout the period. For the middle instrument, there are differences of up to $\pm 30°$, reflecting anomalies in the instrument heading data apparent in panel (b). Prior to circa 02:00 on 7[th] February 2015, $\Phi$ is in close agreement with the other instruments. The burst mean heading, $\langle \phi_H \rangle$, exhibits a steady clockwise rotation, then reduces by $\sim 60°$ between bursts and remains fairly constant over a $2\,\mathrm{hour}$ period (7 bursts), at the end of which it jumps by $\sim 90°$ and reverts to tracking the rotating tide. During this hiatus, both $U$ and $\Phi$ are in excellent agreement with

the other instruments, but the subsequent jump in $\langle\phi_H\rangle$ introduces an offset of $\sim -30°$ in $\Phi$. Approximately $4\,\text{hours}$ later, the offset changes sign over a period of $\sim 1\,\text{hour}$, the transition coinciding with $\langle\phi_H\rangle$ progressing through $360°/0°$. The offset subsequently changes sign again as $\langle\phi_H\rangle$ increases past $180°$ and again when it next progresses through $360°/0°$. A second sudden change in $\langle\phi_H\rangle$ between bursts occurs at circa 20:00 the same day, just prior to the second transition through $360°/0°$, but affects just a single burst.

The incidence of such events was rare, with no clear periodicity apparent, albeit mostly occurring when $U$ was low during neap tides, suggesting the possibility of a mechanical cause. However, the coincidence of the change in sign of the offset in $\Phi$ with the progression of $\langle\phi_H\rangle$ through $180°$ and $360°/0°$ suggests the possibility of a compass sensor problem. Despite this issue affecting the calculation of the earth coordinate current direction for some bursts, there is no indication of any problems with the variation of $\phi_H$ during a burst.

Panel (b) shows that the variation in $\phi_H$ was limited during the majority of bursts. However, in each of two successive burst at circa 20:00 on 7[th] February, the lower instrument completes an anticlockwise rotation over a period of $\sim 90\,\text{s}$, with the heading then returning to a similar value to that prior to the rotation. Over the rest of the burst, the heading varies over a range $\sim 30°$ as in other bursts. The events coincide with $U$ being at a minimum and the direction of rotation is opposite to the rotation of the tide, suggesting the effect may be due to a relaxation of accumulated tension in the mooring.

Panels (c) to (e) show the time series of $\phi_H$, $\phi_P$ and $\phi_R$ for the individual burst identified by the green box in panel (b). The plots show that the instruments all oscillate throughout the period of the burst, the frequency and amplitude of the oscillation varying between instruments. The range and frequency of these oscillations are examined further in the following sections and in Appendix A.

### 3.3 Heading Variation

For each ADCP and deployment period, the instrument heading, $\phi_H$, typically oscillated around a burst mean that rotated with the tide. For each burst, the heading data was analysed as the burst maximum heading range, $\Delta\phi_H$, evaluated as the absolute difference between the minimum and maximum $\phi_H$ expressed on a continuous basis, such that if the instrument completes a full rotation during the burst $\Delta\phi_H \geqslant 360°$; and the number of heading oscillations per burst, $n_{\phi_H}$, evaluated as the number of times $\phi_H$ increased above the burst mean heading, $\langle\phi_H\rangle$, such that $\phi_H - \langle\phi_H\rangle$ changed from negative to positive.

Statistics for each instrument and deployment period are included in Appendix A. The middle instrument, mounted in an open frame, exhibited the largest amplitude oscillations, with $\Delta\phi_H \geqslant 180°$ in more than $9\,\%$ of bursts during the "autumn" deployment period 3 (22[nd] August to 20[th] November 2014) and approximately $7\,\%$ of bursts during the "winter" deployment period 4, compared with $1\,\%$ to $2\,\%$ for the upper and lower instruments. The middle instrument was also typically subject to more oscillations per burst than the other instruments. The lower instrument typically exhibited the fewest and smallest amplitude oscillations.

Figure 5 illustrates the variation of $\Delta\phi_H$ and $n_{\phi_H}$ with the concurrent tidal current speed, $U$, and spectral significant wave height, $H_{m_0}$, for the "winter" deployment period 4. $U$ is the current speed from the burst mean horizontal earth coordinate velocity components, depth averaged across the reliably resolved bin levels. $H_{m_0}$ is calculated from the Triaxys buoy data as

per equation (9) and interpolated to the ADCP observation times. Bursts are aggregated based on $U$ and $H_{m_0}$ for $0\,\mathrm{m\,s^{-1}} \leqslant U \leqslant 0.7\,\mathrm{m\,s^{-1}}$ and $0\,\mathrm{m} \leqslant H_{m_0} \leqslant 12\,\mathrm{m}$ with aggregation bin sizes $\delta U = 0.0175\,\mathrm{m\,s^{-1}}$ and $\delta H_{m_0} = 0.3\,\mathrm{m}$. The left, centre and right columns show the data for the upper, middle and lower ADCP respectively.

Panels (a) to (c) show the mean of the maximum heading range, $\overline{\Delta\phi_H}$, for the bursts in each $(\delta U, \delta H_{m_0})$ aggregation bin; panels (d) to (f) the mean number of heading oscillations, $\overline{n_{\phi_H}}$; and panels (g) to (i) the percentage of bursts in each bin. Plots for the other deployment periods (not shown) demonstrate the same basic patterns, subject to the more limited $H_{m_0}$ range.

For all instruments, $\overline{\Delta\phi_H}$ is highest when $U$ is low, tending to decrease with increasing $U$. There is also evidence of $\overline{\Delta\phi_H}$ increasing with $H_{m_0}$, most clearly for the middle instrument. Conversely, $\overline{n_{\phi_H}}$, exhibits a clear tendency to increase with $U$ for all instruments, but is relatively insensitive to variations in $H_{m_0}$. The rate at which $\overline{n_{\phi_H}}$ increases with $U$ varies between the instruments, but they all exhibit the same basic response.

The variation from a few large oscillations at low $U$ to an increasing number of smaller amplitude oscillations at higher $U$ is consistent with the oscillations being primarily a hydraulic response. The relatively higher values of $\overline{\Delta\phi_H}$ and $\overline{n_{\phi_H}}$ for the middle instrument suggests that the open frame housing is more susceptible to motion than the spherical housing used for the other instruments.

## 3.4 Tilt Variation

The pitch and roll data for each profile was used to compute the tilted beam angle relative to the vertical, $\alpha_i$ (°), for each beam $i$, as described in Appendix A. Figure 6 illustrates the dependence of beam tilt on concurrent $U$ and $H_{m_0}$ during the "winter" deployment period 4. Mean values are again taken across bursts aggregated in $(\delta U, \delta H_{m_0})$ bins, where $\delta U$ is $0.0175\,\mathrm{m\,s^{-1}}$ and $\delta H_{m_0}$ is $0.3$ m. Data for the upper, middle and lower instruments are shown in the left, centre and right columns respectively.

Panels (a) to (c) show the mean absolute burst tilt across all beams, $\overline{\delta\langle\alpha\rangle}$, where $\delta\langle\alpha\rangle = \underline{|\langle\alpha_i\rangle - \theta|}$, with $\langle\alpha_i\rangle$ being the burst mean tilt for beam $i$, the vertical bars indicating the absolute value and the underline indicating the mean across the beams. Panels (d) to (f) show the mean of the beam tilt variation range, $\overline{\Delta\alpha}$, with $\Delta\alpha$ being the mean across the beams of the difference between the burst maximum and minimum $\alpha_i$ values for beam $i$. Panels (g) to (i) show the mean beam tilt oscillations per burst, $\overline{n_\alpha}$, where $n_\alpha$ is the mean across the beams of $n_{\alpha_i}$ which is evaluated as the number of times the sign of $\alpha_i - \langle\alpha_i\rangle$ changes from negative to positive during the burst. The plots for other deployment periods (not shown) are similar, subject to the more limited $H_{m_0}$ range.

The mean beam tilt angle, $\overline{\delta\langle\alpha\rangle}$, exhibits a clear dependence on $U$, increasing with increasing $U$ for all instruments, the effect being relatively weaker for the upper instrument and strengthening with instrument depth. The mean beam tilt angle inevitably understates the tilt for individual beams e.g. for the lower instrument $\overline{\delta\langle\alpha\rangle} \geqslant 10°$ for just $0.6\%$ of bursts during deployment 4, although $4.6\%$ of bursts had at least one beam with that level of tilt. In such circumstances the opposing beams will differ significantly in their orientation to the prevailing current, as well as spanning different vertical ranges.

The mean burst tilt range, $\overline{\Delta\alpha}$, clearly increases with increasing $H_{m_0}$, suggesting that the range of the rocking motion about the tilt axes is primarily driven by the surface wave forced orbital motion. This is consistent with the upper buoy on the mooring rising and falling with the wave, thereby varying the vertical angle of the mooring. Some tendency for $\overline{\Delta\alpha}$ to increase with

increasing $U$ is also apparent for the middle instrument and, to a lesser extent, the upper instrument. Large ranges are observed for both the upper and middle instrument, with the mean across the beams exceeding $20°$ in $0.3\%$ of bursts and at least one beam exceeding $20°$ in $1.3\%$ of bursts for the middle instrument during this deployment period, the equivalent figures for the upper instrument being $0.2\%$ and $1.0\%$ respectively. The beam tilt range is significantly reduced for the lower instrument, consistent with $\Delta\alpha$ being influenced by surface waves.

The variation in the mean beam tilt oscillation frequency, as indicated by $\overline{n_\alpha}$, is relatively limited. The highest values affecting the middle and lower instruments and occurring at low $H_{m_0}$ but with no consistent trends.

## 4  Retained Bias in Synthesised Sheared Flow

The observations demonstrate that tethered ADCP may be subject to both a mean tilt due to drag on the mooring, as well as significant oscillatory variation in both heading and tilt over the period of an observation burst. In the presence of a sheared flow, this motion will unavoidably result in a proportion of the non-turbulent velocity difference between bins being retained in $b'$, contributing to the structure function and biasing the $\varepsilon$ estimates derived using the standard regression method.

This retained bias was investigated using synthesised velocities for a range of scenarios with the ADCP subject to oscillatory variations in heading, pitch and roll. For each scenario, along-beam velocities, $b$, were synthesised for a burst of observations following the procedure detailed in Appendix B. The ADCP geometry was based on the TRDI Workhorse ADCP, with a default beam angle $\theta = 20°$ and a vertical bin size $\delta z = 0.1\,\mathrm{m}$, with bin 1 centred at $\delta z_1 = 1\,\mathrm{m}$ and 30 bins per beam. The default observation burst comprised 300 profiles at $1\,\mathrm{Hz}$.

The residual velocity, $b'$, was calculated by deducting the burst mean, $b' = b - \langle b \rangle$, and the second-order longitudinal structure function, $D_{LL}(i,j,r_n)$, evaluated as per equation (2) using a bin-centred difference scheme for each beam $i$, bin $j$ and all possible bin separation distances, $r_n$, based on multiples of the along-beam bin size $\delta r = \delta z / \cos\theta$ (Wiles et al., 2006). TKE dissipation rate values, $\varepsilon^s$, were calculated using the standard regression method of a least-squares linear regression to equation (4) with $r_{\max} = 2.02\,\mathrm{m}$ (equivalent to a maximum separation of 19 bins) and including the single bin separation and the equation (5) constant $C_2 = 2.0$, with the superscript indicating that the values are from synthesised data. The depth-average, $\varepsilon_i^s$ for beam $i$, was taken as the mean across bins 11 to 20 for which all $r_n \leqslant r_{\max}$ were evaluated.

No turbulence was introduced in either the along-beam velocities or the structure function, therefore $\varepsilon_i^s$ was the retained bias due to the motion of the ADCP.

$\varepsilon_i^s$ values were normalised as a proportion of the potential bias, $\varepsilon_{45}^b$, calculated from the along-beam velocity, $b$, for the same background flow and ADCP configuration, with the ADCP vertical, static and oriented with the heading at $45°$ to the background flow direction, such that each beam has the same difference angle to the flow and therefore the same potential bias.

The default background flow was specified with a speed at the ADCP transducer head $U_0 = 0.25\,\mathrm{m\,s^{-1}}$; with depth constant direction (to) $\beta = 90\,°\mathrm{N}$; shear $S^2 = 1 \times 10^{-4}\,\mathrm{s^{-2}}$; and no surface waves. Testing confirmed that the results were insensitive to $U_0$ and that both $\varepsilon_i^s$ and $\varepsilon_{45}^b$ scaled as $S^3$, such that $\varepsilon_i^s / \varepsilon_{45}^b$ was independent of $S^2$.

## 4.1 Heading Variation Example

Figure 7 illustrates the impact of heading oscillation for an example scenario. Initial heading $\phi_H(0)$ and mean current direction $\beta$ are both $90\,°$N. The heading oscillation range $\Delta\phi_H = 60°$ and the period $t_{\phi_H} = 30\,$s; and the instrument is vertical, with $\phi_P(t)$ and $\phi_R(t)$ zero for all $t$.

Panel (a) shows the variation in the synthesised ADCP bin positions in an $[x, y, z]$ coordinate framework referenced to the transducer head, see Appendix B. The sweep of each of the beams is shown by the shaded areas, with lines indicating the positions for bins $1, 5, 10, 15 \ldots 30$ and markers for the bin 30 centre position at times $t = 0\,$s (circle), $2\,$s (square), $8\,$s (diamond) and $22\,$s (triangle).

The first $30\,$s of the synthesised along-beam velocity time series for bin 16 in each beam $i$, $b_i(16, t)$ (m s$^{-1}$), is shown in panel (b), the variation repeating over the $300\,$s duration of the burst. Beam 1 (blue line) is initially oriented across the background flow, such that $b_1(16, 0)$ is zero ($t = 0\,$s, circle marker). As the heading changes, beam 1 initially points increasingly upstream (square marker at $t = 2\,$s) and $b_1(16, t)$ varies with the sine of the heading difference angle, reaching a positive maximum at $t_{\phi_H}/4$ when $\phi_H(t) = \phi_H(0) + \Delta\phi_H/2$ (just before the diamond marker at $t = 8\,$s. The heading then rotates back towards the mean position and $b_1(16, t)$ reduces to zero at $t_{\phi_H}/2$. As the oscillation continues, $b_1(16, t)$ reaches a maximum negative at $3t_{\phi_H}/4$ (close to the triangle marker at $t = 22\,$s) and returns to zero at $t_{\phi_H}$, with the oscillation repeating until the end of the burst. Since the ADCP is vertical, symmetry means that $b_2(16, t)$ (red line) has the same magnitude but opposite sign to $b_1(16, t)$ for all $t$.

Beam 3 (yellow line) is initially oriented directly downstream, so that $b_3(16, 0)$ has a maximum negative value. As the heading changes, the magnitude of $b_3(16, t)$ reduces as the cosine of the heading difference angle, reaching a minimum at $t_{\phi_H}/4$ then increasing to regain its maximum value at $t_{\phi_H}/2$; the variation repeating over the second half of the oscillation period. Compared with $b_1(16, t)$, $b_3(16, t)$ varies with double the oscillation frequency but a much smaller amplitude and has a non-zero mean. Symmetry again means that the $b_4(16, t)$ (purple line) has the same magnitude as $b_3(16, t)$ but opposite sign.

Since the burst mean for beams 1 and 2 is approximately zero, the periodic variation in $b$ is fully retained in $b'$, including any velocity differences between bins due to the sheared flow. Conversely, for beams 3 and 4, the variation in $b$ is greatly reduced, so that the majority of velocity difference between bins is not retained in $b'$. This is reflected in panel (c), which shows the time series for $\delta b'$ for bin 16 with $r_n = r_{max}$ ($19\delta r$) for each beam, $\delta b'_i(16, 19\delta r)$ (mm s$^{-1}$). The opposing beams in each beam pair have identical values but opposite sign, whilst the magnitude of the oscillation for beams 1 and 2 is clearly much larger than that for beams 3 and 4.

Panel (d) shows the time series for the squared velocity difference $[\delta b'_i(16, 19\delta r)]^2$ (mm$^2$ s$^{-2}$), which is positive for all $t$. Values for the opposing beam pairs are identical, with the burst mean for beams 1 and 2 (red line overlying blue line) clearly significantly larger than that for beams 3 and 4 (purple line overlying yellow line).

Panel (e) shows the structure function $D_{LL}(i, 16)$ for each beam and a range of $r_n$ values, including $r_{max}$ indicated by the vertical green line, plotted against $r^{\frac{2}{3}}$, demonstrating both the marked difference between the beam pairs and the non-linear growth of $D_{LL}$ with $r^{\frac{2}{3}}$. Again, beams 1 (solid blue line) and 2 (red bullet markers) are identical, as are beams 3 (solid yellow

line) and 4 (purple bullet markers). The dotted blue (beam 1) and yellow (beam 3) lines indicate the linear regression fit for all $r_n \leqslant r_{\max}$ with no restriction on the regression intercept.

The annotation in panel (e) shows the normalised residual bias $\varepsilon_i^s / \varepsilon_{45}^b$ for each beam, indicating the retained fraction of the potential bias in each beam. For this scenario the residual bias arises almost exclusively in beams 1 and 2, which have a mean alignment across the current direction and are only exposed to the current by the oscillation, whilst the contribution from beams 3 and 4, which are closely aligned with the current direction, is negligible.

## 4.2 Heading Variation Scenarios

The potential impact of the heading varying was evaluated across scenarios with $\phi_H(0)$ varied in $5°$ increments over the range $30°$N to $150°$N; $\Delta\phi_H$ varied in $10°$ increments over the range $0°$ to $450°$; and $18\ t_{\phi_H}$ options over the range $10\,\text{s}$ to $360\,\text{s}$ with the ADCP vertical for all scenarios i.e. $\phi_P(t)$ and $\phi_R(t)$ being $0°$ for all $t$, yielding 20,275 scenarios. The ranges for $\Delta\phi_H$ and $t_{\phi_H}$ were chosen taking account of the variation in the observations described in section'3 and with the aim of encompassing the likely range of impacts.

The results are summarised in Figure 8. Panel (a) shows the variation of the beam averaged normalised residual bias, $\underline{\varepsilon}^s / \varepsilon_{45}^b$, the underline indicating the mean of $\varepsilon_i^s$ across the four beams, with the difference angle between the initial ADCP heading and the background flow direction, $\psi = \beta - \phi_H(0)$, for selected heading oscillation ranges, $\Delta\phi_H$, and a fixed heading oscillation period $t_{\phi_H}$ of $30\,\text{s}$. Since the heading oscillates around $\phi_H(0)$, the burst mean heading $\langle\phi_H\rangle \approx \phi_H(0)$, with any slight difference arising from the burst period not being an exact multiple of the oscillation period. Hence $\psi$ is also the burst mean heading offset angle relative to the background flow.

For each $\Delta\phi_H$, there is a limited variation in $\underline{\varepsilon}^s / \varepsilon_{45}^b$ with $\psi$, being highest when $\psi = 0°$ and lowest for $\psi = \pm 45°$, the ratio between the minimum and the maximum decreasing with increasing $\Delta\phi_H$. This variation is superimposed on the clear trend for $\underline{\varepsilon}^s / \varepsilon_{45}^b$ to increase with $\Delta\phi_H$, as indicated by comparing the lines for the selected options. This is illustrated further in panel (b), which shows the mean $\underline{\varepsilon}^s / \varepsilon_{45}^b$ (black line), together with the $25\,\%$ to $75\,\%$ (dark grey shading) and $5\,\%$ to $95\,\%$ (light grey shading) ranges for scenarios aggregated on the basis of $\Delta\phi_H$, combining scenarios with the various $\psi$ and $t_{\phi_H}$ options. Mean $\underline{\varepsilon}^s / \varepsilon_{45}^b$ is negligible for $\Delta\phi_H \leqslant 50°$, then increases to reach a maximum of $\sim 1$ at $\Delta\phi_H \sim 270°$, before declining gradually to $\sim 0.8$ as $\Delta\phi_H$ continues to increase.

For each $\Delta\phi_H$, the range of $\underline{\varepsilon}^s / \varepsilon_{45}^b$ is limited, confirming the limited impact of $\psi$ and $t_{\phi_H}$ on the beam mean residual bias. However, this masks a much greater variation in the normalised residual bias for individual beams, $\varepsilon_i^s / \varepsilon_{45}^b$, as illustrated in panel (c), which shows the mean (black line) and the $25\,\%$ to $75\,\%$ (dark grey shading) and $5\,\%$ to $95\,\%$ (light grey shading) ranges for $\varepsilon_i^s / \varepsilon_{45}^b$ aggregated by $\Delta\phi_H$. The potential variation between beams increases markedly over the range $50° \leqslant \Delta\phi_H \leqslant 200°$ before reducing, with maximum $\varepsilon_i^s / \varepsilon_{45}^b$ values exceeding $1.5$ occurring for $180° \leqslant \Delta\phi_H \leqslant 260°$.

The vertical lines in panels (b) and (c) of Figure 8 indicate the deployment 4 median (dotted line) and $90^{\text{th}}$ percentile $\Delta\phi_H$ values for the upper (grey) and middle (black) instruments fromt he observations described in section 3. The results suggest that for these observations, the proportion of the potential bias likely to be retained is typically low, although under some circumstances it might exceed $50\,\%$ for the middle instrument.

## 4.3 Tilt Variation Example

Figure 9 illustrates the impact of oscillation on the pitch tilt axis for a sample scenario with constant heading and no roll, with all panels as described in Figure 7. The initial pitch angle $\phi_P(0) = 0°$, the oscillation range $\Delta\phi_P = 20°$ and the oscillation period $t_{\phi_P} = 30\,\mathrm{s}$. The heading is constant and aligned with the background flow and there is no tilt on the roll axis i.e. $\phi_H(t) = \beta$ and $\phi_R(t) = 0°$ for all $t$.

Panel (a) shows the sweep of the beams. At $t = 0\,\mathrm{s}$ (circle marker), $\phi_P$ is $0°$ and the instrument is vertical, such that beams 1 and 2 (blue and red) are oriented normal to the current and their along-beam velocities are zero. As $\phi_P(t)$ becomes positive, beam 3 (yellow) is tilted towards the vertical, so that its bins are higher in the water column than those in beam 4 (purple), as indicated by the position of square markers for the bin 30 positions after $2\,\mathrm{s}$. This tilts beams 1 and 2 slightly upstream and $b$ becomes positive for all bins in both beams (red line overlying blue line), increasing to a positive maximum at $t_{\phi_P}/4$

when $\phi_P(t) = \Delta\phi_P/2$ (just prior to the diamond marker at $t = 8\,\mathrm{s}$), then reducing as $\phi_P$ declines, so that both are zero again at $t_{\phi_P}/2$, as shown in panel (b). As $\phi_P$ becomes negative, beams 1 and 2 are both tilted slightly downstream and $b$ becomes negative, reaching a maximum negative value at $3t_{\phi_P}/4$ when $\phi_P(t) = -\Delta\phi_P/2$ (close to the triangle marker at $t = 22\,\mathrm{s}$), before returning to zero after a full oscillation period. Consequently, for beams 1 and 2, $b$ is the same, oscillating in phase between positive and negative values with period $t_{\phi_P}$ and with the burst mean $\langle b_i \rangle \approx 0\,\mathrm{m\,s^{-1}}$.

Beams 3 and 4 (yellow and purple) initially have a symmetrical orientation downstream and upstream respectively, such that for any bin, $b_3(0) = -b_4(0)$. As $\phi_P$ becomes positive, the change in the relative orientation of beam 3 to the horizontal current reduces the magnitude of the along-beam velocity component $|b_3|$, as shown in panel (b), despite the change in the bin depths increasing the local current speed. In contrast, beam 4 is tilted towards the horizontal, the change in orientation resulting in $|b_4|$ increasing, despite the reduction in the local current speed at the new bin depths. As the pitch oscillation continues, $b_3$ and $b_4$

vary in phase with each other, with $\langle b_i \rangle \approx b_i(0)$.

     The slight differences in the depth ranges of the beams result in slight differences in $\delta b_i'$ between the beams, as can be seen in panel (c). Whilst the variation is identical for beams 1 and 2 (red line overlying blue line), the $|\delta b_3'|$ maximum during the positive $\phi_P$ phase of the oscillation is larger than during the negative $\phi_P$ phase of the oscillation, with the situation reversed for beam 4. This is clearer in panel (d), which shows $[\delta b_i']^2$. Beams 1 and 2 are identical, with the largest maxima and identical

values during both the positive and negative $\phi_P$ phases, whilst the maxima for beams 3 and 4 are lower and differ between the phases, such that the beam 3 values are larger during the positive $\phi_P$ phases and the beam 4 values during the negative $\phi_P$ phases.

     The differences between beams 3 and 4 during the positive and negative phases of the oscillation are symmetric, therefore the burst mean values used by the $D_{LL}$ are identical, as shown in panel (e). Beams 3 and 4 yield identical results with $\varepsilon_i^s/\varepsilon_{45}^b$

values approximately $30\,\%$ lower than those for beams 1 and 2, for which the normalised residual bias as a result of the ADCP motion is $\sim 0.1$.

     Oscillation about the roll axis, which in this scenario is oriented along the background flow, has no impact on $b_i$ for beams 1 and 2 which remain normal to the flow throughout the burst. The roll oscillation has a minimal impact on the vertical observa-

tion range for beams 3 and 4 resulting in a normalised residual bias in these beams of $\mathcal{O}10^{-6}$, highlighting the significance of the instrument orientation to the background flow on the impact of oscillation around the individual tilt axes.

## 4.4 Tilt Variation Scenarios

The potential impact of pitch and roll oscillations was evaluated for a sample of $500\,000$ scenarios based on the default configuration and sheared background flow. For each scenario, a constant heading angle was specified with $\phi_H(0)$ selected at random (equal probability for each option) between $0°$ to $355°$ at $5°$ intervals and $\Delta\phi_H = 0°$. Initial pitch angle, $\phi_P(0)$, was randomly selected from the range $-10°$ to $10°$ at $1°$ intervals; pitch oscillation range, $\Delta\phi_P$, from the range $-20°$ to $20°$ at $1°$ intervals, with the sign indicating the initial rotation direction; and the pitch oscillation period, $t_{\phi_P}$, randomly selected in the range $10\,\text{s}$ to $70\,\text{s}$. The initial roll angle, $\phi_R(0)$, roll oscillation range, $\Delta\phi_R$, and roll oscillation period, $t_{\phi_R}$ were randomly selected from the same ranges as the pitch equivalents, whilst the roll phase offset, $\delta t_{\phi_R}$, was randomly selected in the range $0\,\text{s}$ to $30\,\text{s}$. The ranges for each variable were chosen based on the sensor ranges and the observations described in section 3, with the aim of covering the likely potential impacts.

Figure 10 shows the mean (black line), $25\%$ to $75\%$ range (dark grey shading) and $5\%$ to $95\%$ range (light grey shading) for the beam averaged normalised residual bias $\underline{\varepsilon}^s/\varepsilon_{45}^b$, together with the 95th percentile (dotted line with triangle markers) and maximum (grey line with square markers) individual beam normalised residual bias $\varepsilon_i^s/\varepsilon_{45}^b$ for scenarios aggregated by: panel (a) the heading offset angle to the background flow, $\psi$; panel (b) the sum of the absolute values of the initial tilt angles, $|\phi_P(0)| + |\phi_R(0)|$; and panel (c) the sum of the absolute values of the tilt oscillation ranges, $|\Delta\phi_P| + |\Delta\phi_R|$.

Panel (a) illustrates how the symmetry of the ADCP beam geometry is reflected in the impact of the instrument orientation relative to the background flow. The mean $\underline{\varepsilon}^s/\varepsilon_{45}^b$ is effectively constant (mean $0.023$) across all $\psi$, whilst the range of the beam average values (light grey shading) is largest when the heading is such that one of the beams is aligned with the background flow i.e. $\psi$ is $0°$, $\pm90°$ or $\pm180°$, and smallest when all beams are at $45°$ to the background flow i.e. $\psi$ is $\pm45°$ or $\pm135°$.

There is a marked contrast between the 95th percentile of the individual beam normalised residual bias $\varepsilon_i^s/\varepsilon_{45}^b$ (dotted line with triangle marker), which closely tracks that of the beam averaged values, and the beam maximum (grey line with square marker). They vary in anti-phase, with maximum $\varepsilon_i^s/\varepsilon_{45}^b$ values of $\sim 0.3$ occurring with $\psi \sim \pm45°$ or $\pm135°$.

Panel (b) shows that the mean and range of $\underline{\varepsilon}^s/\varepsilon_{45}^b$ exhibit minimal dependence on the mean tilt, as indicated by the sum of the initial tilt angles $|\phi_P(0)| + |\phi_R(0)|$, again recognising that the specified tilt oscillation means that $\langle\phi_P\rangle \approx \phi_P(0)$ and $\langle\phi_R\rangle \approx \phi_R(0)$. The $5\%$ to $95\%$ range actually narrowing slightly as the mean tilt increases. The 95th percentile of the individual beam $\varepsilon_i^s/\varepsilon_{45}^b$ values is also effectively constant, whilst there is a gradual increase in the maximum $\varepsilon_i^s/\varepsilon_{45}^b$ values as $|\phi_P(0)| + |\phi_R(0)|$ increases from $0°$ to $6°$, above which it is relatively constant.

Panel (c) indicates that for the scenarios examined, the residual bias is primarily determined by the total absolute oscillation range, $|\Delta\phi_P| + |\Delta\phi_R|$. The mean $\underline{\varepsilon}^s/\varepsilon_{45}^b$ is $\sim 0$ for $|\Delta\phi_P| + |\Delta\phi_R| \leqslant 15°$, gradually increasing to a maximum of $\sim 0.09$ (black line). The range of $\underline{\varepsilon}^s/\varepsilon_{45}^b$ values is narrow for all $|\Delta\phi_P| + |\Delta\phi_R|$ options. The 95th percentile of the individual beam $\varepsilon_i^s/\varepsilon_{45}^b$ values closely tracks that of $\underline{\varepsilon}^s/\varepsilon_{45}^b$ for $|\Delta\phi_P| + |\Delta\phi_R| \leqslant 20°$, above which is increases at a slightly higher rate. This is also

reflected in the beam maximum $\varepsilon_i^s/\varepsilon_{45}^b$ values, which grows at an increasing rate, exceeding $0.3$ for the extreme scenarios with $|\Delta\phi_P| + |\Delta\phi_R|$ approaching $40°$.

The vertical lines in panel (c) are the deployment 4 median (dotted line) and $90^{\text{th}}$ percentile (solid line) $\Delta\phi_P + \Delta\phi_R$ values for the upper (grey) and middle (black) instruments from the observations described in section 3. The results suggest that for these observations, oscillation on the tilt axes is unlikely to result in the beam average retaining a significant fraction of the potential bias, although for individual beams it may exceed $10\,\%$ in some circumstances.

### 4.5   Effectiveness of the Modified Regression Method

Scannell et al. (2017) identified that the orbital velocities due to surface waves contribute a periodic velocity component that varies between bins due to their spatial separation, leading to residual non-turbulent velocity differences in the structure function and biased $\varepsilon$ estimates. Over a limited spatial range, the velocity difference between bins varies linearly with separation distance, resulting in a contribution to the second-order structure function with a $r^2$ length-scale dependency.

As described in section 2.2, any residual structure function contribution due to the motion of the ADCP in the presence of linear shear will also exhibit an $r^2$ length-scale dependency. This suggests that the modified regression including terms for both $r^{\frac{2}{3}}$ and $r^{\frac{2}{3}})^3$, as per equation (6), should also be effective in isolating any non-turbulent contribution due to shear from the genuine turbulence signal.

This was tested on the synthesised data (with or without the deduction of the burst mean) and was found to completely eliminate the bias, yielding $\varepsilon$ values of $\mathcal{O}\,10^{-30}\ \mathrm{W\,kg^{-1}}$ or less, reflecting the numerical precision of the calculations.

The synthesised data is a pure "bias" signal and therefore optimised to be identified and isolated. The effectiveness of the modified method with real observations affected by ADCP motion in a sheared flow is likely to be determined by the noise in the signal and the choice of $r_{\mathrm{max}}$. Furthermore, since the same term in the modified regression is used to isolate both the bias contribution due to surface waves and that due to residual shear, it isn't possible to distinguish between these factors in the interpreting the impact of applying the modified regression to real observations when both may be relevant.

## 5   Discussion

The standard structure function methodology assumes that the along-beam velocities observed by an ADCP can be decomposed into a component due to the background flow and the time-varying turbulent velocities required to calculate $\varepsilon$. Deducting the mean or linear trend over a burst of observations for each bin therefore removes the component due to the background flow, including any non-turbulent velocity differences between bins due to shear. For this assumption to be valid, there must be no spatially-varying periodic non-turbulent velocity contribution to the observed velocity, such as that due to surface waves or, as considered here, due to the motion of the ADCP in a sheared background flow.

If the orientation of the ADCP varies, the burst mean velocity in any bin unavoidably underestimates the background flow contribution in some profiles and overestimates it in others. If the background flow is sheared, the residual velocity when the

burst mean or linear trend is deducted will include a proportion of the associated non-turbulent velocity differences between bins.

The potential contribution to the second-order structure function if the velocity differences due to linear shear in the background flow were wholly retained in the residual velocity is here shown to scale with the square of both the shear and the separation distance, see equation (8). The potential bias will therefore scale as the cube of the shear and will be sensitive to both the choice of the maximum separation distance over which the structure function is evaluated and the ADCP bin size (which determines the number of resolved separation distances).

Data from long-term deployments of three ADCP mounted inline on a buoyancy-tensioned mooring demonstrates the instruments oscillating in heading, pitch and roll. The heading variation was found to vary between fewer, larger amplitude oscillations when the background flow is slowest and a higher number of smaller amplitude oscillation as the background flow speed increased. The background flow speed also directly influenced the mean tilt angle for the instruments as the drag determines the shape of the mooring. Surface waves had some influence on heading variation, however the impact was most 495 apparent in the range of the tilt oscillation. There was also evidence that the way in which the ADCP was mounted influenced the movement, with the instruments in spherical syntactic buoys subject to less motion than that in an open frame.

    Synthesised along-beam velocity data based on a standard TRDI Workhorse ADCP geometry was used to evaluate the impact of instrument motion in a linearly sheared flow. The residual bias was normalised by the potential bias for the defined geometry, background flow and with all beams having the same relative orientation to the flow.

Based on a wide range of synthesised scenarios, the normalised residual bias was found to be primarily determined by the oscillation angular range, both for heading and instrument tilt.

    Testing indicated that the normalised residual bias becomes increasingly significant for heading angular oscillation ranges exceeding $50°$, with the possibility of the full potential bias being retained in one or more beams if the angular range exceeded $140°$. The frequency of occurrence of heading oscillations exceeding $50°$ in the observations examined was dependent on 505 the instrument mounting, but affected more than $50\%$ of observations for the instrument mounted in an open frame during some deployments. Furthermore, since the heading oscillation was unconstrained, angular variations of over $360°$ occasionally occurred.

    Oscillation on the tilt axes is inherently constrained by the tension in the mooring, therefore the potential angular range is limited. The synthesised scenarios suggest that the beam-averaged normalised residual bias due to tilt oscillation will reach 510 $10\%$ only under exceptional circumstances. However, the maximum residual bias for an individual beam, which increases with the total of the pitch and roll angular ranges, can reach $30\%$ of the potential bias under exceptional circumstances.

    The velocity difference between bins due to shear, retained in the along-beam velocity as a consequence of the ADCP motion, vary linearly with separation distance. This is consistent with that arising from the spatial gradient of the orbital velocities forced by surface gravity waves, suggesting that the modified regression in equation (6), as described by Scannell 515 et al. (2017), should be effective in isolating the turbulence signal from any bias. This was confirmed when the modified regression was applied to the synthesised scenarios described in section 4, with the potential bias being completely eliminated. The results suggest that the modified regression may be useful in a wider range of circumstances than removing bias due to

surface waves, isolating all non-turbulent velocity differences that scale linearly with separation distance, although without distinguishing between possible sources.

This analysis suggests that under most circumstances the motion of a tethered ADCP is unlikely to be a significant source of errors in $\varepsilon$ estimates derived using the standard structure function methodology. However, since the potential bias scales with the cube of the shear and depends on factors such as the bin size and the length scale over which the structure function is evaluated, there may be circumstances in which it is significant. Furthermore, since the level of retained bias is dependent on the motion of the ADCP, it is relevant to identify this as an issue for consideration both as part of the deployment planning and

of the data quality assurance and analysis. The following suggestions may therefore be of interest to other researchers:

1. *Mooring design:* Mounting the ADCP in a streamlined buoy designed to maintain a fixed orientation relative to the background current is recommended for all deployments on a tethered mooring. If that isn't an option, mounting the ADCP in a spherical buoy is likely to result in less motion than using an open frame.

2. *ADCP configuration:* Ensure that the instrument orientation sensors (heading, pitch and roll) are working properly and
that the instrument is configured to save the data at the same temporal resolution as the velocity profiles.

3. *Initial QA:* Check for periodic variations in heading, pitch or roll to determine whether the ADCP was subject to significant motion during the observation bursts. In particular, evaluate the heading angular variation range $\Delta\phi_H$, with $\Delta\phi_H \geqslant 50°$ suggested as a threshold above which the possibility of bias should be considered.

4. *Initial QA:* Check for periodic variation in the along-beam velocity data collected. One option is to examine the burst
variance of the along-beam velocity and check for any monotonic trend in the variance between bins, which may indicate a non-turbulent contribution and potential cause of bias.

5. *Shear:* Convert the along-beam velocity data to earth coordinates and determine the level of shear. This can be used to determine the maximum potential bias by computing the sheared structure function $D_{LL}^b$ as per equation (8) based on the bin size and beam angle and then calculating the potential bias $\varepsilon^b$ for the proposed $r_{\max}$.

6. *Structure function QA:* Check for non-linearity of $D_{LL}$ versus $r^{2/3}$. This is perhaps most easily achieved by examining the sensitivity of $\varepsilon$ to increasing $r_{\max}$, with an increasing trend probably indicating a non-turbulent contribution to $D_{LL}$ and therefore a bias in the calculated $\varepsilon$ values.

7. *Structure function QA:* Test whether $\varepsilon$ values are more independent of $r_{\max}$ when using the modified regression equation (6). If so, this suggests a non-turbulent contribution to $D_{LL}$ but care should be taken to determine the source of the
non-turbulent contribution and verify that the associated velocity difference between bins varies linearly with separation distance before assuming the modified method is applicable.

*Data availability.* The ADCP data referred to in Section 3 is currently being prepared for submission to British Oceanographic Data Centre.

**Appendix A: Celtic Sea Turbulence Mooring**

The moorings were deployed at a site in the central Celtic Sea, latitude $49°24'$ N, longitude $8°36'$ W. The site has a nominal
depth of $145\,\mathrm{m}$, is more than $200\,\mathrm{km}$ from any coast and over $125\,\mathrm{km}$ from the shelf break.

The overall deployment period was from late March 2014 to late July 2015, during which time it was serviced four times, with the interval between recovery and re-deployment varying between 1 and 7 days. The same instruments were used for each period, in the same mooring arrangement and with the same sampling configuration. Table A1 shows the dates for the individual deployment periods together with the associated number of observation bursts returned by each instrument.

**A1  Heading**

Table A2 provides information on the heading variation for each instrument and each deployment, together with the number of observation bursts, $n_{obs}$, and the mean depth, $\overline{z}\,(\mathrm{m})$. The burst maximum heading variation, $\Delta\phi_H$, is evaluated as the absolute difference between the minimum and maximum $\phi_H$ expressed on a continuous basis, such that if the instrument completes a full rotation during the burst $\Delta\phi_H \geqslant 360°$. The table shows $\overline{\Delta\phi_H}$, being the mean $\Delta\phi_H$ for the instrument over the deployment
period, together with the $10^{\text{th}}$, $50^{\text{th}}$ and $90^{\text{th}}$ percentile values and the percentage of bursts for which $\Delta\phi_H \geqslant 360°$.

The number of heading oscillations per burst, $n_{\phi_H}$, was evaluated as the number of times $\phi_H$ increased above the burst mean heading, $\langle\phi_H\rangle$, such that $\phi_H - \langle\phi_H\rangle$ changed from negative to positive. The table shows $\overline{n_{\phi_H}}$, being the mean across all bursts; together with the percentage of bursts for which $n_{\phi_H} \leqslant 1$; and the $50^{\text{th}}$ and $90^{\text{th}}$ percentile $n_{\phi_H}$ values.

Examination of a sample of bursts for which $n_{\phi_H} \leqslant 1$ indicated that they were characterised by a significant step change in
the heading, resulting in two distinct sub-periods during the burst. Despite $\phi_H$ oscillating about the relevant mean during each sub-period, there was just a single crossing of $\langle\phi_H\rangle$, resulting in $n_{\phi_H}$ being 0 or 1.

The heading variation is further illustrated in Figure A1. Panels (a) - (c) show the cumulative distribution of $\Delta\phi_H$, with the line colour indicating the deployment as per the legend in panel (a) and the embedded table in each panel showing the percentage of bursts per deployment when $\Delta\phi_H \geqslant 180°$ and $360°$. Panels (d) - (f) show the cumulative distribution of $n_{\phi_H}$,
with the embedded tables showing the percentage of bursts per deployment when $n_{\phi_H} \geqslant 50$. Panels (a) and (d) relate to the upper instrument; (b) and (e) the middle instrument; and (c) and (f) the lower instrument.

The middle instrument is subject to significantly higher levels of heading variation, both in terms of the the range of the angular variation and the number of oscillations per burst. This is interpreted as being a consequence of the different housing used for the instruments in the mooring - the upper and lower instruments being embedded within a spherical syntactic buoy,
whilst the middle instrument was in an open frame.

The same housings were used for each deployment, so the differences in the $\Delta\phi_H$ and $n_{\phi_H}$ distributions between deployments for the individual instruments must arise either from performance differences of mooring elements e.g. swivels or wires, or from differing environmental conditions.

## A2 Tilt

Pitch and roll, $\phi_P$ and $\phi_R$, typically have a constant sign throughout an observation burst, with the burst mean values $\langle\phi_P\rangle$ and $\langle\phi_R\rangle$ tending to have a consistent sign throughout a deployment. This indicates that the initial orientation of the instruments in the mooring results in a preferred orientation relative to the plane of the mooring, which persists throughout the deployment with limited variation, despite the rotation of the mooring with the tide.

Tables A3 and A4 provide summary statistics for the pitch and roll data for each instrument during each of the deployments.

For the absolute burst mean tilts, $|\langle\phi_P\rangle|$ and $|\langle\phi_R\rangle|$, the tables show the deployment mean, $\overline{|\langle\phi_P\rangle|}$ and $\overline{|\langle\phi_R\rangle|}$, together with the percentage of bursts $\geqslant 5°$ and $\geqslant 10°$. For the burst oscillation ranges, $\Delta\phi_P$ and $\Delta\phi_R$, and the burst oscillation counts, $n_{\phi_P}$ and $n_{\phi_R}$, the tables show the deployment mean together with the $10^{\text{th}}$, $50^{\text{th}}$ and $90^{\text{th}}$ percentiles. $\Delta\phi_P$ and $\Delta\phi_R$ being evaluated as the absolute difference between the burst minimum and maximum $\phi_P$ and $\phi_R$ respectively; and $n_{\phi_P}$ and $n_{\phi_R}$ being evaluated as the number of instances during a burst when the tilt increases through the burst mean e.g. when $\phi_P - \langle\phi_P\rangle$ changes from

negative to positive.

The non-zero values for $\overline{|\langle\phi_P\rangle|}$ and $\overline{|\langle\phi_R\rangle|}$ suggest that the instruments were typically tilted from the vertical during a burst. The percentage of bursts with high $|\langle\phi_P\rangle|$ or $|\langle\phi_R\rangle|$ tends to be highest for the lower instrument and lowest for the upper instrument, consistent with the mooring exhibiting a catenary shape due to lateral loading.

The deployment mean burst ranges, $\overline{\Delta\phi_P}$ and $\overline{\Delta\phi_R}$ tend to decline with instrument depth and to vary in a consistent manner

between deployments, being highest during the "autumn" and "winter" deployments 3 and 4 and lowest during the "summer" deployment 2.

The oscillation frequency, as indicated by $\overline{n_{\phi_P}}$ and $\overline{n_{\phi_R}}$, is consistent across all instruments and deployments, being higher than the equivalent $\overline{n_{\phi_H}}$, particularly for the upper and lower instruments.

In order to evaluate the combined impact of pitch and roll, the tilted beam angle relative to the vertical, $\alpha_i$ for beam $i$, was

600 calculated for each beam, following Woodgate and Holroyd (2011). The true pitch, $\phi_{P_t}$, correcting for the influence of roll, is first calculated from the observed pitch and roll as (Teledyne RD Instruments, 2014):

$$\phi_{P_t} = \arctan(\tan(\phi_P)\cos(\phi_R)) \tag{A1}$$

The tilted beam angle relative to the vertical, $\alpha_i$ for beam $i$, is then:

$$\cos(\alpha_1) = -\sin(\phi_R)\sin(\theta) + \cos(\theta)\sqrt{1 - \sin^2(\phi_R) - \sin^2(\phi_{P_t})}$$

$$\cos(\alpha_2) = \sin(\phi_R)\sin(\theta) + \cos(\theta)\sqrt{1 - \sin^2(\phi_R) - \sin^2(\phi_{P_t})}$$

$$\cos(\alpha_3) = \sin(\phi_{P_t})\sin(\theta) + \cos(\theta)\sqrt{1 - \sin^2(\phi_R) - \sin^2(\phi_{P_t})}$$

$$\cos(\alpha_4) = -\sin(\phi_{P_t})\sin(\theta) + \cos(\theta)\sqrt{1 - \sin^2(\phi_R) - \sin^2(\phi_{P_t})} \tag{A2}$$

where $\theta$ is the instrument beam angle, $20°$ or $30°$ as appropriate and $\phi_{P_t} = -\phi_{P_t}$ if the instrument is downward-facing (Woodgate and Holroyd, 2011).

The variation in the ADCP beam average tilt for each of the instruments during each deployment is illustrated in Figure A2. Panels (a) to (c) show the cumulative probability of $\Delta\alpha$ for each instrument (column) and deployment (line colour), where $\Delta\alpha$ is mean across the four beams of the difference between the maximum and minimum $\alpha_i$ values for beam $i$ during a burst. Panels (d) to (f) show the cumulative probability of $n_\alpha$ for each instrument and deployment, where $n_\alpha$ is the mean across the four beams of $n_{\alpha_i}$, being the number of times that the sign of $a_i - \langle\alpha_i\rangle$ changes from negative to positive during a burst.

There is a broadly consistent seasonal pattern in the distributions for all three instruments. The "spring" deployment periods 1 and 5 (blue and green lines respectively) are similar, with $\Delta\alpha$ typically increasing for the "autumn" deployment period 3 (yellow line), highest for the "winter" deployment period 4 (purple line) and lowest for the "summer" deployment period 2 (red line), which is consistent with the variation in wave energy conditions. The tables inset in each panel show the percentage of bursts for each deployment when the mean tilt range is $\geqslant 5°$ and $10°$, confirming that the tilt range for the upper and middle

instruments is significantly more than than for the lower instrument.

The distributions of $n_\alpha$ suggest that there is only limited variation between instruments and deployments. The middle and lower instruments exhibit a wider range of $n_\alpha$, although the median is $\sim 30$ oscillations per burst for all instruments and deployments.

## Appendix B: Synthesised Along-Beam Velocity

The along-beam velocities observed by an ADCP, $b$, may include contributions due to the potentially sheared background flow, $\overline{b}$; the orbital motion forced by surface gravity waves, $\widetilde{b}$; and turbulent motions, $b'$. We assume that these combine linearly as:

$$b = \overline{b} + \widetilde{b} + b' \tag{B1}$$

Along-beam velocity $b_{i,n}(t)$ for beam $i$, bin $n$ at time $t$, are therefore synthesised by first determining the instantaneous ADCP bin position $\mathbf{x}_{i,n}(t) = [x_{i,n}(t), y_{i,n}(t), z_{i,n}(t)]$. We then compute the earth coordinate velocities at that location due

to the background flow, $\overline{\mathbf{u}}_{i,n}(t) = [\overline{u}_{i,n}(t), \overline{v}_{i,n}(t), \overline{w}_{i,n}(t)]$, and surface waves, $\widetilde{\mathbf{u}}_{i,n}(t) = [\widetilde{u}_{i,n}(t), \widetilde{v}_{i,n}(t), \widetilde{w}_{i,n}(t)]$. Finally, we determine $b_{i,n}(t)$ as the along-beam component of $\mathbf{u}_{i,n}(t) = \overline{\mathbf{u}}_{i,n}(t) + \widetilde{\mathbf{u}}_{i,n}(t)$ in the rotated beam coordinates and assuming that there is no turbulence such that $b'$ is zero. Note that synthesised velocities are calculated directly from the specified background flow and any surface waves, without any allowance for observational noise or the spatial and temporal averaging that will affect actual observations to differing degrees depending on the operating mode.

All scenarios were based on a Teledyne RDI Workhorse ADCP beam geometry, with a four beam Janus-style convex transducer head, such that with the instrument vertical, all beams have the same angle to the vertical, $\theta$ (°); with heading angle, $\phi_H$ (°N), indicating the compass direction of the horizontal projection of beam 3; pitch, $\phi_P$ (°), indicating the rotation in the plane of beams 3 and 4; and roll, $\phi_R$ (°), indicating rotation in plane of beams 1 and 2, with both $\phi_P$ and $\phi_R$ being 0° indicating that the ADCP is vertical and the convention for direction of rotation being as per Teledyne RD Instruments (2010), taking

account of whether the ADCP is specified as upward- or downward-facing.

A standard burst configuration of 300 profiles collected at $1\,\text{Hz}$ was adopted, with 30 bins per beam and a default vertical bin size of $\delta z = 0.1\,\text{m}$ with bin 1 centred at $\delta z_1 = 1.0\,\text{m}$.

## B1 Bin Positions

Bin positions were calculated in Cartesian coordinates relative to the ADCP transducer head, with the $x$-axis oriented due East, the $y$-axis due North and the $z$-axis pointing vertically upward, such that the transducer head is at $[x, y, z] = [0, 0, 0]$.

Unit vectors describing the orientation of each beam with $\phi_H$, $\phi_P$ and $\phi_R$ all 0° and the ADCP upward-facing are then:

$$\begin{bmatrix} \widehat{\mathbf{X}}_1 & \widehat{\mathbf{X}}_2 & \widehat{\mathbf{X}}_3 & \widehat{\mathbf{X}}_4 \end{bmatrix} = \begin{bmatrix} \sin\theta & -\sin\theta & 0 & 0 \\ 0 & 0 & \sin\theta & -\sin\theta \\ \cos\theta & \cos\theta & \cos\theta & \cos\theta \end{bmatrix} \tag{B2}$$

and the along-beam bin centre position for all bins, common to all beams, is:

$$R = \left(\delta z_1 + \delta z [0\ 1\ 2 \ldots N-1]\right) / \cos\theta \tag{B3}$$

where $N$ is the number of bins. The non-rotated coordinates for all of the bins in beam $i$ are then given by $\mathbf{X}_i = \widehat{\mathbf{X}}_i R$, as illustrated in panel (a) of Figure A3.

Heading variation was prescribed as a sinusoidal oscillation, with an initial angle, $\phi_H(0)$ (°N), an oscillation angular range, $\Delta\phi_H$ (°), and a heading oscillation period, $t_{\phi_H}$ (s), such that at profile time $t$:

$$\phi_H(t) = \phi_H(0) + (\Delta\phi_H/2)\sin\left(2\pi t/t_{\phi_H}\right) \tag{B4}$$

with $t$ varying from $0\,\mathrm{s}$ to $299\,\mathrm{s}$ over the burst and $\Delta\phi_H = 0°$ or $t_{\phi_H} = 0\,\mathrm{s}$ indicating a constant heading. Similarly the pitch variation over the burst was defined as:

$$\phi_P(t) = \phi_P(0) + (\Delta\phi_P/2)\sin\left(2\pi t/t_{\phi_P}\right) \tag{B5}$$

and the roll variation as:

$$\phi_R(t) = \phi_R(0) + (\Delta\phi_R/2)\sin\left(2\pi(t + \delta t_R)/t_{\phi_R}\right) \tag{B6}$$

with the only difference being the option to additionally specify $\delta t_R$ as a phase offset.

Bin positions at time $t$ are then determined by rotation about the appropriate axes, with $\phi_H$ describing rotation about the $z$-axis, $\phi_P$ the $x$-axis and $\phi_R$ the $y$-axis as:

$$M_{\phi_H}(t) = \begin{bmatrix} \cos\phi_H(t) & \sin\phi_H(t) & 0 \\ -\sin\phi_H(t) & \cos\phi_H(t) & 0 \\ 0 & 0 & 1 \end{bmatrix}$$

$$M_{\phi_P}(t) = \begin{bmatrix} 1 & 0 & 0 \\ 0 & \cos\phi_P(t) & -\sin\phi_P(t) \\ 0 & \sin\phi_P(t) & \cos\phi_P(t) \end{bmatrix}$$

$$M_{\phi_R}(t) = \begin{bmatrix} \cos\phi_R(t) & 0 & \sin\phi_R(t) \\ 0 & 1 & 0 \\ -\sin\phi_R(t) & 0 & \cos\phi_R(t) \end{bmatrix} \tag{B7}$$

subject only to $\phi_R = \phi_R + 180°$ if the ADCP is specified as downward-facing. The positions for all bins in beam $i$ are then given by:

$$\mathbf{X_i}(t) = \begin{bmatrix} X_i(t) \\ Y_i(t) \\ Z_i(t) \end{bmatrix} = M_{\phi_H}(t)\, M_{\phi_P}(t)\, M_{\phi_R}(t)\, \widehat{\mathbf{X}_i}\, R \tag{B8}$$

## B2  Velocity due to the Background Flow

A steady horizontal current, $\overline{\mathbf{u}}$, is defined with a speed at the transducer head depth, $\overline{u}_0$ $(\mathrm{m\,s^{-1}})$; compass direction (to), $\beta$ $(°N)$; and vertical shear-squared, $S^2$ $(\mathrm{s^{-2}})$ with $S$ assumed to be positive such that current speed $\overline{u}$ increases towards the surface and $S^2 = 0\,\mathrm{s^{-2}}$ indicating that the flow velocity is constant over the depth range.

The background flow velocity in earth coordinates at the beam $i$ bin locations for time $t$ is then given by:

$$\overline{\mathbf{U}_i}(t) = \begin{bmatrix} \overline{U}_i(t) \\ \overline{V}_i(t) \\ \overline{W}_i(t) \end{bmatrix} = \begin{bmatrix} \sin\beta\,(\overline{u}_0 + SZ_i(t)) \\ \cos\beta\,(\overline{u}_0 + SZ_i(t)) \\ 0 \end{bmatrix} \tag{B9}$$

with $\overline{U}_i(t)$, $\overline{V}_i(t)$ and $\overline{W}_i(t)$ being the velocity components along the $x$-, $y$- and $z$-axes respectively.

## B3  Orbital Velocity due to Surface Waves

For a monochromatic surface gravity wave, linear wave theory describes the orbital motion as:

$$\widetilde{u} = \frac{gk}{\omega} A_0 \frac{\cosh\big(k(z+h)\big)}{\cosh(kh)} \sin(kx - \omega t)$$

$$\widetilde{w} = -\frac{gk}{\omega} A_0 \frac{\sinh\big(k(z+h)\big)}{\cosh(kh)} \cos(kx - \omega t) \tag{B10}$$

where $\widetilde{u}$ is the velocity component in the direction of wave propagation; $\widetilde{w}$ is the vertical velocity component; $x$ is the distance in the direction of wave propagation; $z$ is depth referenced to the surface and positive upwards; $t$ is time; $g$ is acceleration due to gravity; $k$ is wavenumber given by $k = 2\pi/\lambda$ where $\lambda$ is the wavelength; $A_0$ is the surface amplitude of the wave; and $\omega$ is the radian frequency given by $\omega = ck$ where $c$ is the wave phase speed from the wave dispersion equation:

$$c^2 = \frac{\omega^2}{k^2} = \frac{g}{k} \tanh(kh) \tag{B11}$$

with $h$ being the water column height, such that $z = -h$ at the seabed (Phillips, 1977).

From equation (B10), the wave orbital motion velocity in earth coordinates at the beam $i$ bin locations for time $t$ is given by:

$$\widetilde{\mathbf{U}}_i(t) = \begin{bmatrix} \widetilde{U}_i(t) \\ \widetilde{V}_i(t) \\ \widetilde{W}_i(t) \end{bmatrix} = \begin{bmatrix} \sin\alpha \, \dfrac{gk}{\omega} A_0 \, \dfrac{\cosh\left(k(\widetilde{Z}_i(t)+h)\right)}{\cosh\left(kh\right)} \sin\left(k\widetilde{X}_i(t)-\omega t\right) \\[2ex] \cos\alpha \, \dfrac{gk}{\omega} A_0 \, \dfrac{\cosh\left(k(\widetilde{Z}_i(t)+h)\right)}{\cosh\left(kh\right)} \sin\left(k\widetilde{X}_i(t)-\omega t\right) \\[2ex] -\dfrac{gk}{\omega} A_0 \, \dfrac{\sinh\left(k(\widetilde{Z}_i(t)+h)\right)}{\cosh\left(kh\right)} \cos\left(k\widetilde{X}_i(t)-\omega t\right) \end{bmatrix} \tag{B12}$$

where $\alpha$ (°N) is the wave propagation compass direction (to); $\widetilde{Z}_i(t) = Z_i(t) + z_0$ is the beam $i$ bin depths referenced to the sea surface given an ADCP depth $z_0$; and $\widetilde{X}_i(t)$, is the array of rotated beam $i$ bin positions relative to the direction of wave propagation, calculated as:

$$\widetilde{X}_i = \begin{bmatrix} X_i \\ Y_i \end{bmatrix} \cdot \begin{bmatrix} \sin\alpha \\ \cos\alpha \end{bmatrix} \tag{B13}$$

being the scalar dot product of the horizontal components of the rotated beam bin positions and the horizontal unit vector for the wave propagation direction, as illustrated in panel (b) of Figure A3.

For scenarios including surface waves, the waves were specified in terms of their wavelength, $\lambda$ (m); surface amplitude, $A_0$ (m); and compass direction of propagation (to), $\alpha$ (°N). The depth of the ADCP, $z_0$ (m), was specified within the range $-50\,\mathrm{m} \leqslant z_0 \leqslant -20\,\mathrm{m}$ and a standard water depth of $h = 145\,\mathrm{m}$ was used for all scenarios.

## B4 Along-beam Velocity

The total velocity in earth coordinates at the beam $i$ bin locations at time $t$ is then taken as the linear sum of the velocity due to the background flow and that due to surface waves as:

$$\mathbf{U}_i(t) = \overline{\mathbf{U}}_i(t) + \widetilde{\mathbf{U}}_i(t) \tag{B14}$$

The along-beam velocity for all of the bins in beam $i$, $b_i$, is then calculated as:

$$b_i = -\mathbf{U}_i \cdot \frac{\mathbf{X}_i}{|\mathbf{X}_i|} \tag{B15}$$

being the scalar dot product projection of the total earth coordinate velocity onto the rotated bin position vector $\mathbf{X}_i$, with the negative sign included for consistency with the RDI convention that along-beam velocities are positive towards the transducer.

*Author contributions.* BDS undertook the analysis and prepared the manuscript. Y-DL and TPR commented on and contributed to the development of the manuscript.

*Competing interests.* None of the authors are aware of having any competing interests.

*Acknowledgements.* The observations described in section 3 were collected as part of the United Kingdom (UK) Natural Environment Re-
search Council (NERC) Carbon and Nutrient Dynamics and Fluxes over Shelf Systems (CaNDyFloSS) project (reference NE/K00168X/1),
which forms part of the Shelf Sea Biogeochemistry research programme co-funded by the Department for Environment, Food and Rural
Affairs (Defra). We thank Dr Joanne E. Hopkins of the National Oceanography Centre, Liverpool, the crew of the RRS *Discovery* and the
National Marine Facilities staff for their assistance in collecting the ADCP observations and Jon Turton at the Met Office for supplying
the wave buoy data. BDS was supported by NERC Award 1500369 through the Envision doctoral training program and subsequently, with
Y-DL, by the PEANUTS project (NE/R01275X/1), part of the Changing Arctic Ocean programme funded by NERC and the German Federal
Ministry of Education and Research. We also thank two anonymous reviewers for their detailed comments and suggestions, which have
helped us to improve the manuscript.

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

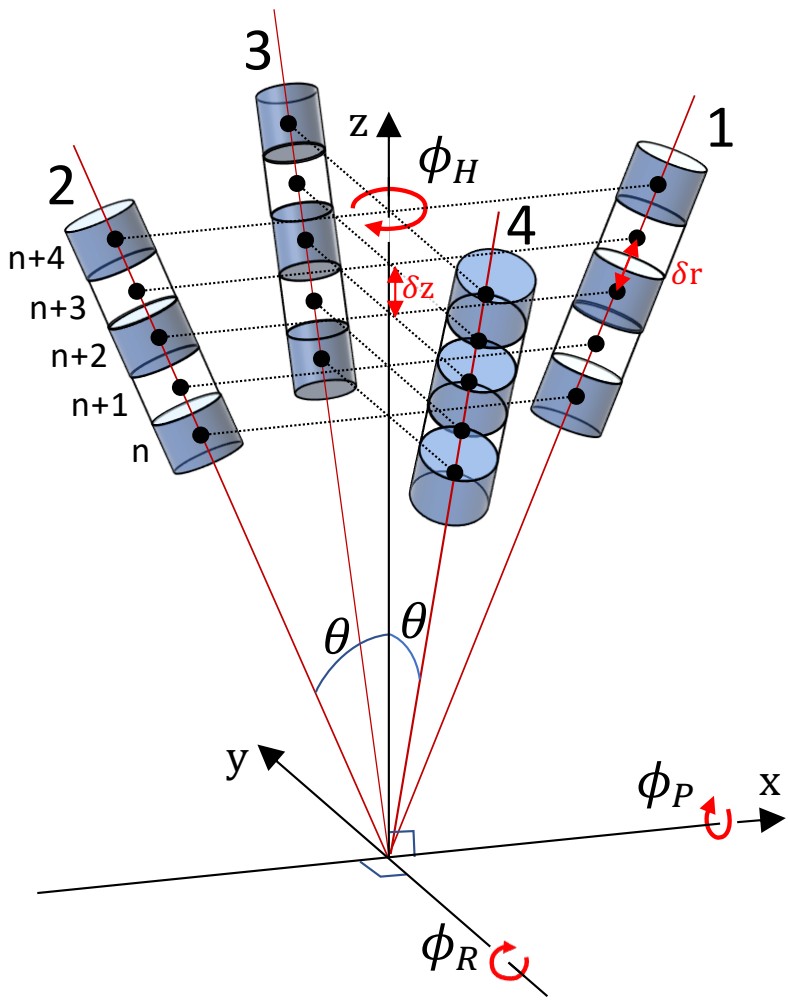

**Figure 1.** Geometry for a Teledyne RDI WorkHorse four-beam ADCP. Solid red lines indicate the centre line for the beams, each with beam angle $\theta$ to the along-instrument $z$-axis (typically oriented vertically), with beams 1 and 2 symmetric about the $z$-axis in the $y = 0$ plane and beams 3 and 4 similarly oriented in the $x = 0$ plane. Bins $n$ to $n + 4$ are shown for each beam, with $\delta z$ being the bin centre separation distance along the $z$-axis and $\delta r = \delta z / \cos \theta$ being the along-beam bin centre separation distance. Heading, $\phi_H$, describes the compass angle for beam 3; pitch, $\phi_P$, the rotation from vertical about the $x$-axis and roll, $\phi_R$, the rotation from vertical about the $y$-axis, with the sign convention dependent on whether the instrument is oriented upwards- or downwards-looking.

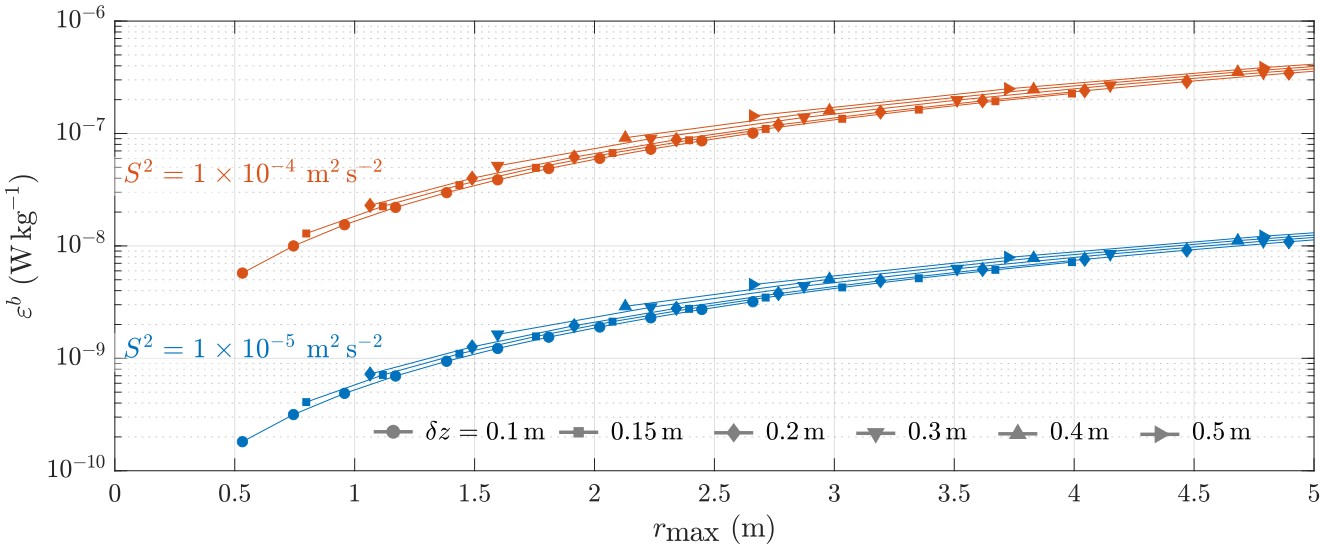

**Figure 2.** Potential bias due to shear using the standard method regression when none of the non-turbulent velocity differences between bins due to shear is removed, $\varepsilon^b$, evaluated for selected levels of shear, $S^2$; bin sizes, $\delta z$; and maximum separation distance used for the regression, $r_{max}$.

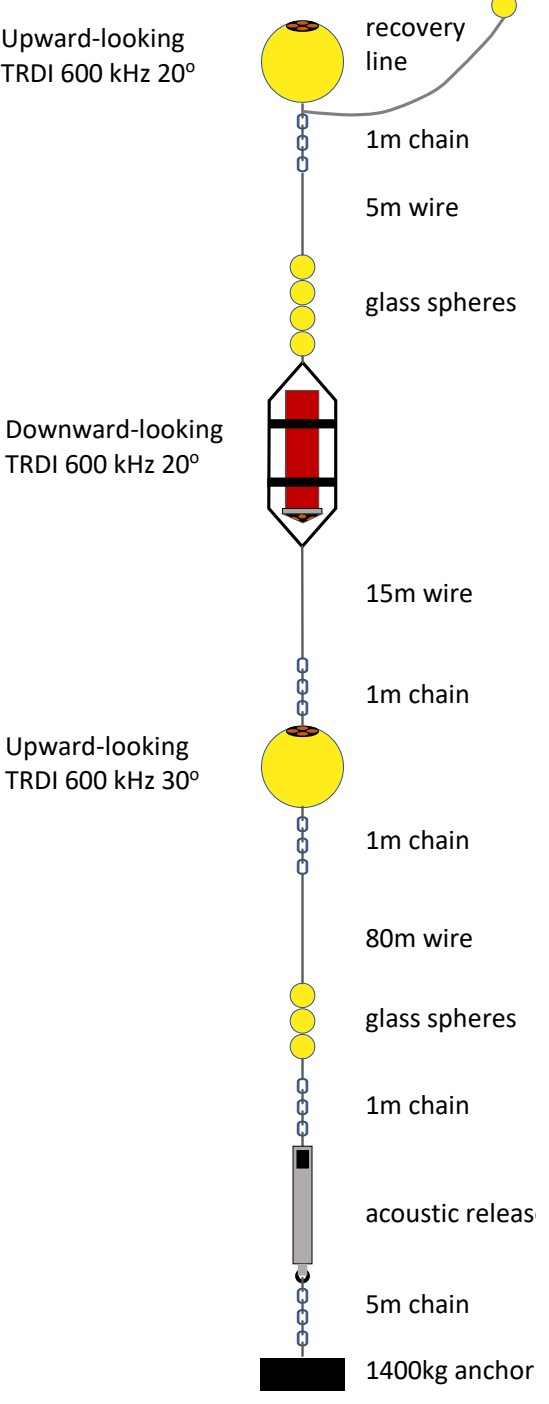

**Figure 3.** Turbulence mooring diagram. *Adapted from figure in the RRS James Cook JC105 cruise report, available from British Oceanographic Data Centre,* `www.bodc.ac.uk`

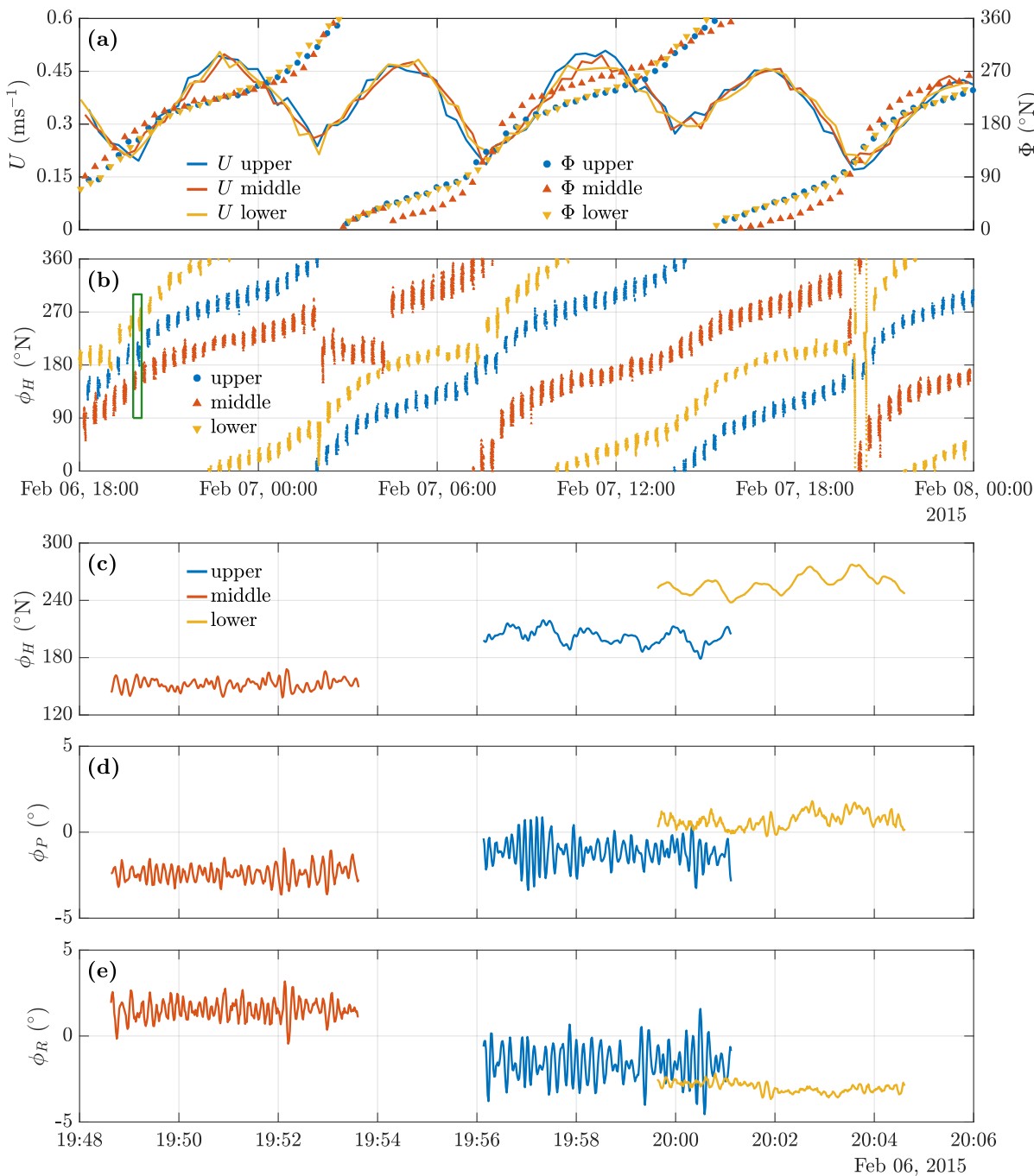

**Figure 4.** Sample heading data. Panel (a) shows the earth coordinate current speed $U$ (solid lines) and direction (to), $\Phi$, (markers), and panel (b) shows the heading, $\phi_H$, for each ADCP (colour). Panels (c) to (e) show the variation in $\phi_H$; pitch, $\phi_P$; and roll, $\phi_R$, respectively for the bursts indicated by the green box in panel (b).

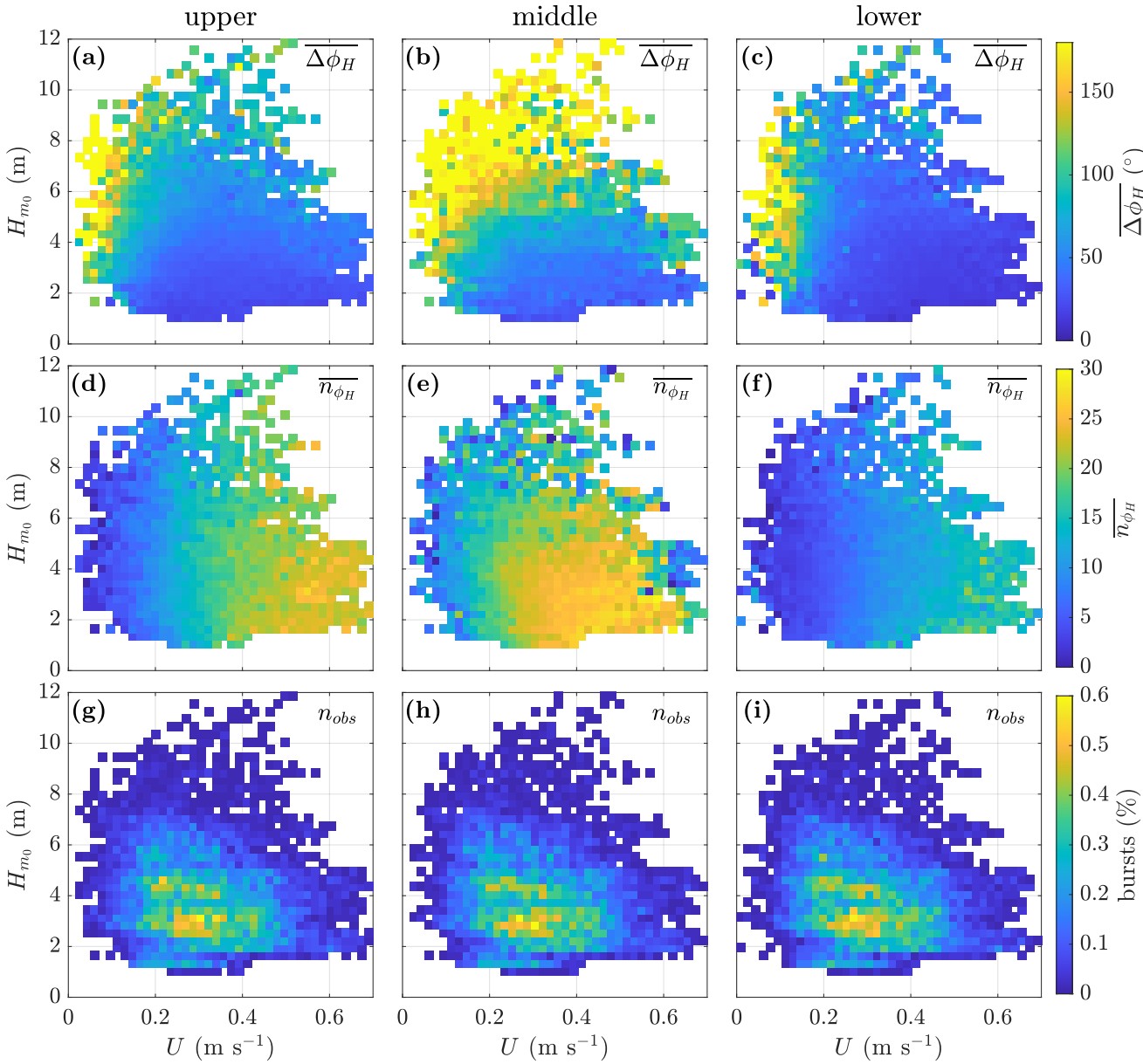

**Figure 5.** Mean heading oscillation range, $\overline{\Delta\phi_H}$ (°) (top row); mean heading oscillations per burst, $\overline{n_{\phi_H}}$ (middle row); and percentage of bursts (bottom row), for bursts aggregated by current speed, $U$ (m s$^{-1}$), and significant wave height, $H_{m_0}$ (m), for deployment period 22 November 2014 to 4 April 2015, with aggregation bin $\delta U = 0.0175 \, \text{m s}^{-1}$ and $\delta H_{m_0} = 0.3 \, \text{m}$. Instrument mooring position is shown above each column.

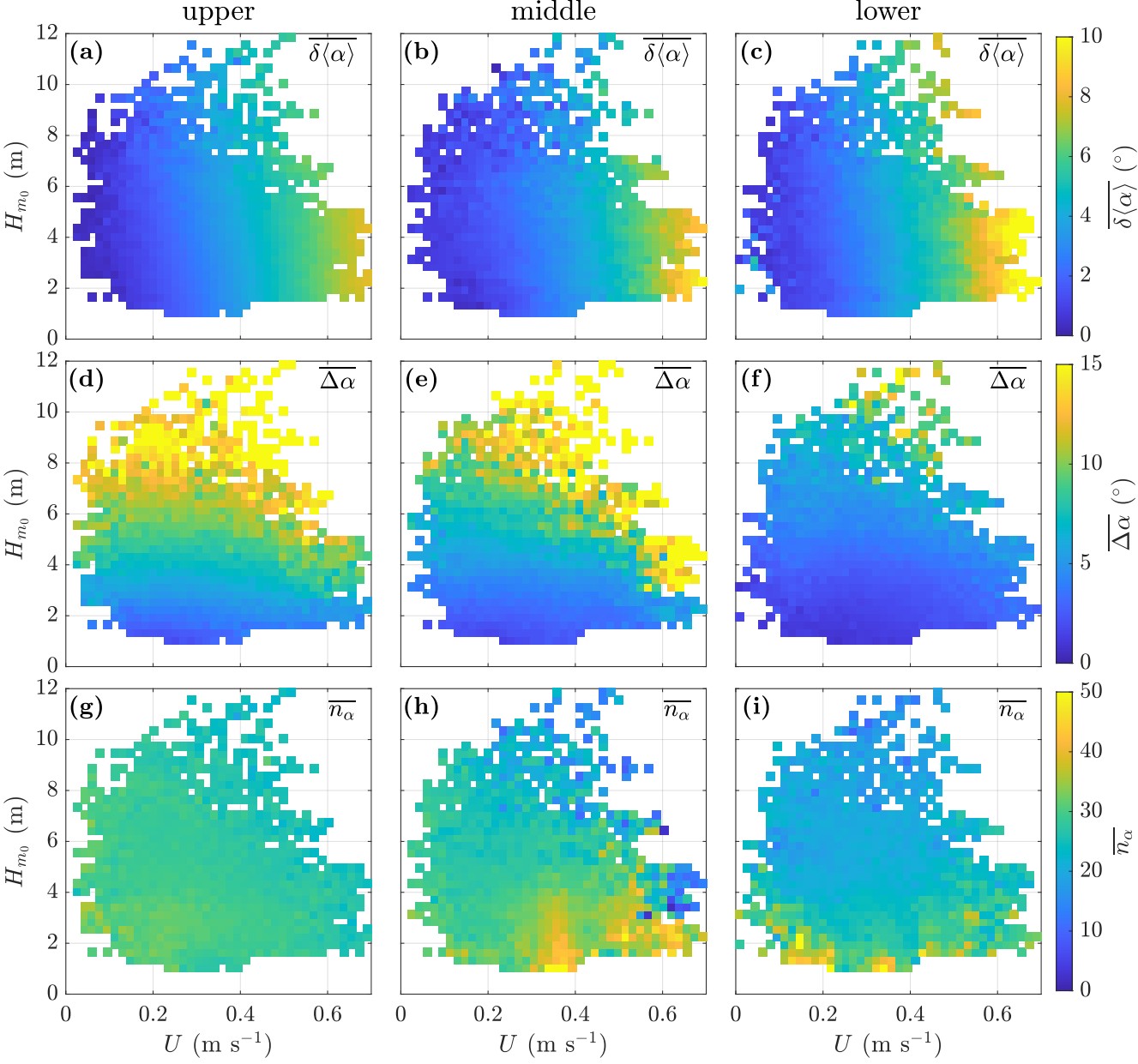

**Figure 6.** Mean tilt, $\overline{\delta\langle\alpha\rangle}$ (° (top row); mean tilt range, $\overline{\Delta\alpha}$ (°) (middle row); and mean tilt oscillations per burst, $\overline{n_{\phi_H}}$ (bottom row), for bursts aggregated by current speed, $U$ (m s$^{-1}$), and significant wave height, $H_{m_0}$ (m), for deployment period 22 November 2014 to 4 April 2015, with aggregation bin $\delta U = 0.0175$ m s$^{-1}$ and $\delta H_{m_0} = 0.3$ m. Instrument mooring position is shown above each column.

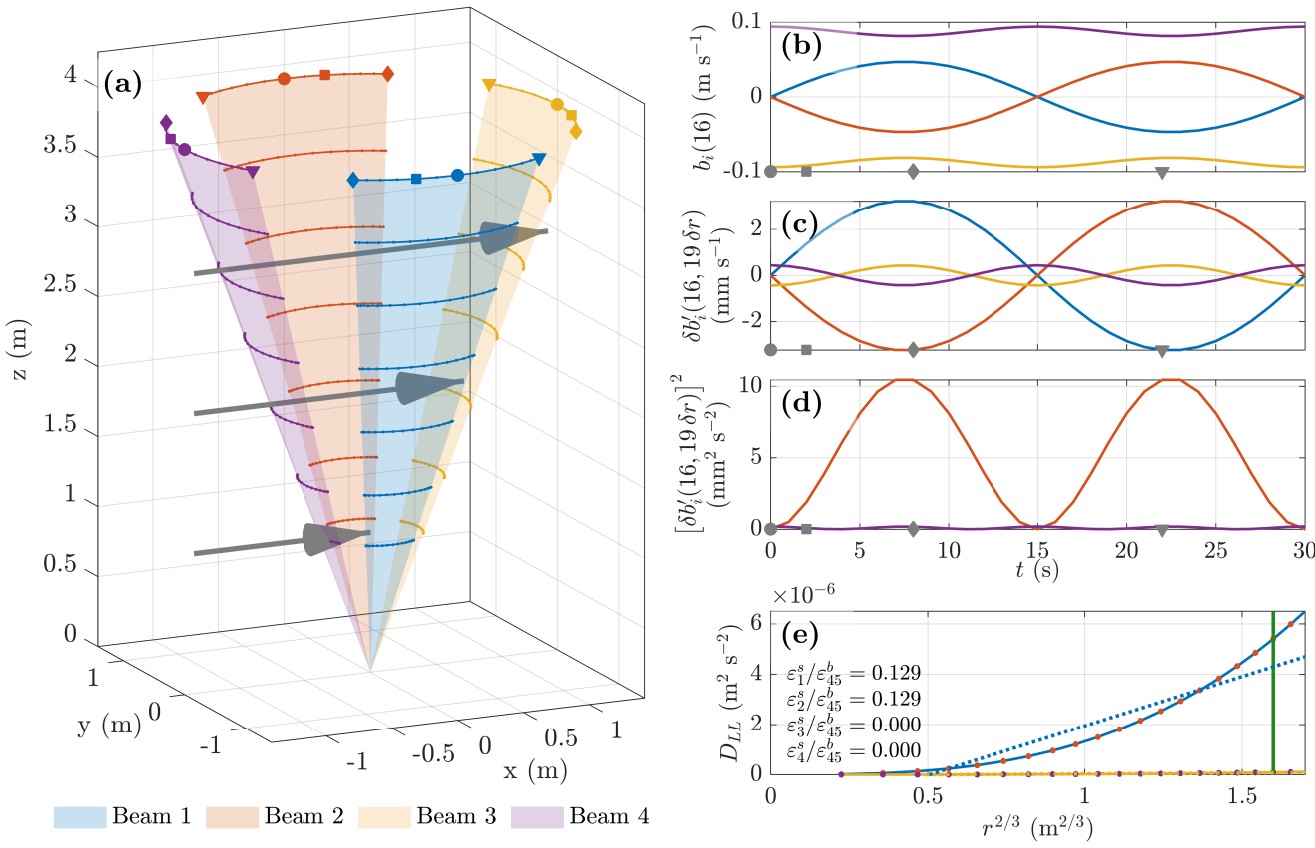

**Figure 7.** Illustration of heading oscillation in a sheared current, with $\phi_H(0) = 90\,°\mathrm{N}$; $\Delta\phi_H = 60°$; $t_{\phi_H} = 30\,\mathrm{s}$; and $\phi_P(t)$ and $\phi_R(t)$ both $0°$ for all $t$. Panel (a) shows the beam sweep with lines indicating the positions of bins $1, 5, 10, 15 \ldots 30$; markers indicate bin 30 centre position at times $t = 0\,\mathrm{s}$ (circle), $2\,\mathrm{s}$ (square), $8\,\mathrm{s}$ (diamond) and $22\,\mathrm{s}$ (triangle); grey arrows indicate the sheared mean current (not to scale). Panels (b) to (d) show the first $30\,\mathrm{s}$ of the $300\,\mathrm{s}$ duration burst time series for bin 16 of each beam (colours as per panel (a) legend) for (b) the along-beam velocity, $b\;(\mathrm{m\,s^{-1}})$; (c) the bin-centred difference of the residual velocity, $\delta b_i'\;(\mathrm{mm\,s^{-1}})$, for $r$ of $19\,\delta r$; and (d) the squared velocity difference, $[\delta b_i']^2\;(\mathrm{mm^2\,s^{-2}})$, with the grey markers in each panel indicating the times of the corresponding shape bin 30 position markers in panel (a). Panel (e) shows the second-order structure function, $D_{LL}\;(\mathrm{m^2\,s^{-2}})$, for beam 1 (blue line), 2 (red bullet), 3 (yellow line) and 4 (purple bullet); with $r_{\max}$ indicated by the vertical green line; and the linear regressions for beams 1 and 3 (dotted lines); with the annotation showing the normalised residual bias $\varepsilon_i^s/\varepsilon_{45}^b$ for each beam.

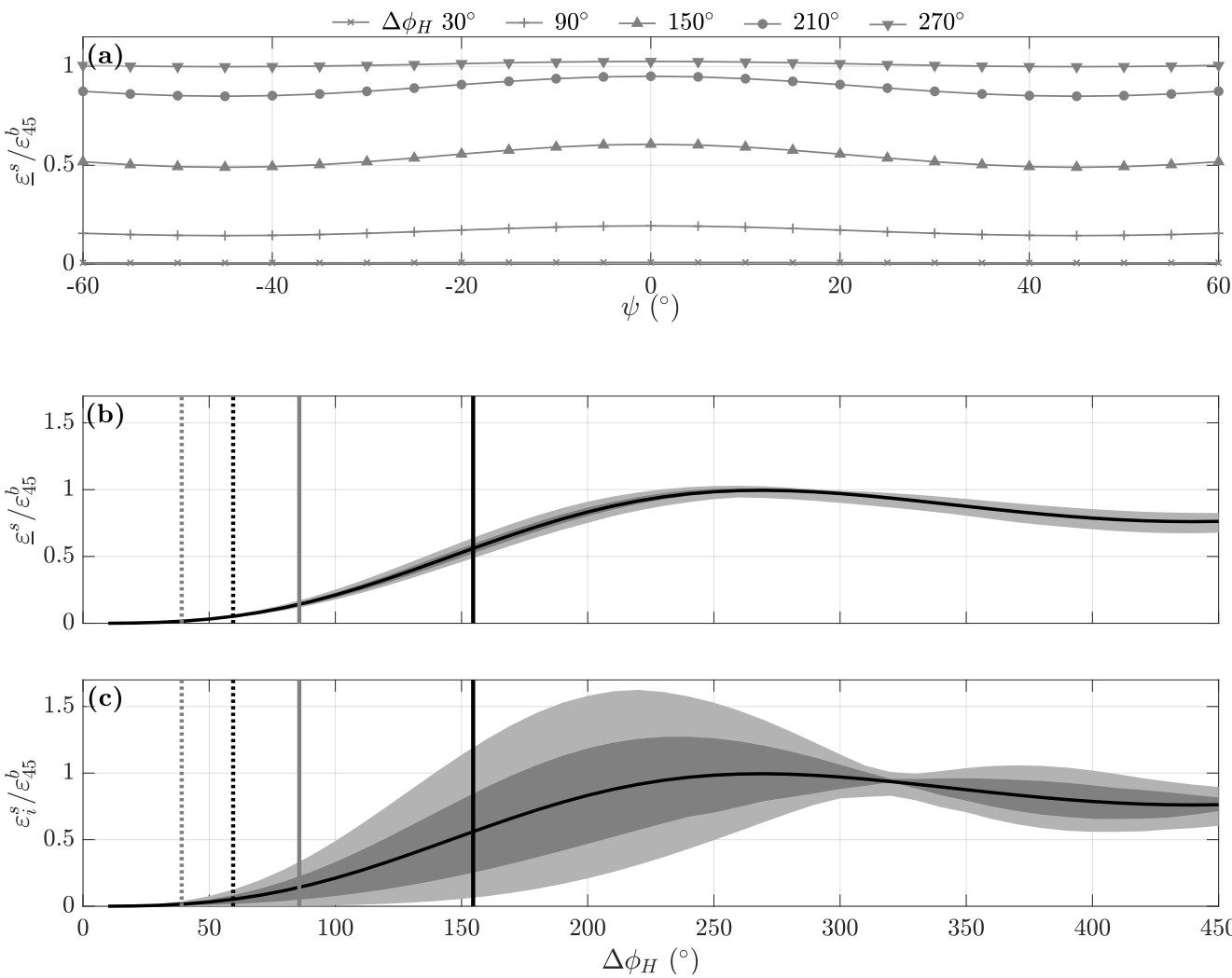

**Figure 8.** Normalised residual bias for heading oscillation scenarios. Panel (a) variation of beam averaged normalised residual bias, $\underline{\varepsilon}^s/\varepsilon_{45}^b$, with initial heading offset angle to the background flow, $\psi$, for selected heading oscillation ranges $\Delta\phi_H$ and fixed heading oscillation period $t_{\phi_H} = 30\,\mathrm{s}$; panel (b) mean $\underline{\varepsilon}^s/\varepsilon_{45}^b$ (black line), with 25 % to 75 % range (dark grey shading) and 5 % to 95 % range (light grey shading) for scenarios aggregated by $\Delta\phi_H$; panel (c) mean and ranges for the maximum normalised beam residual bias $\varepsilon_i^s/\varepsilon_{45}^b$ as in panel (b). The vertical lines in panels (b) and (c) are the deployment 4 median (dotted line) and 90[th] percentile (solid line) $\Delta\phi_H$ values for the upper (grey) and middle (black) instruments from the observations described in section 3.

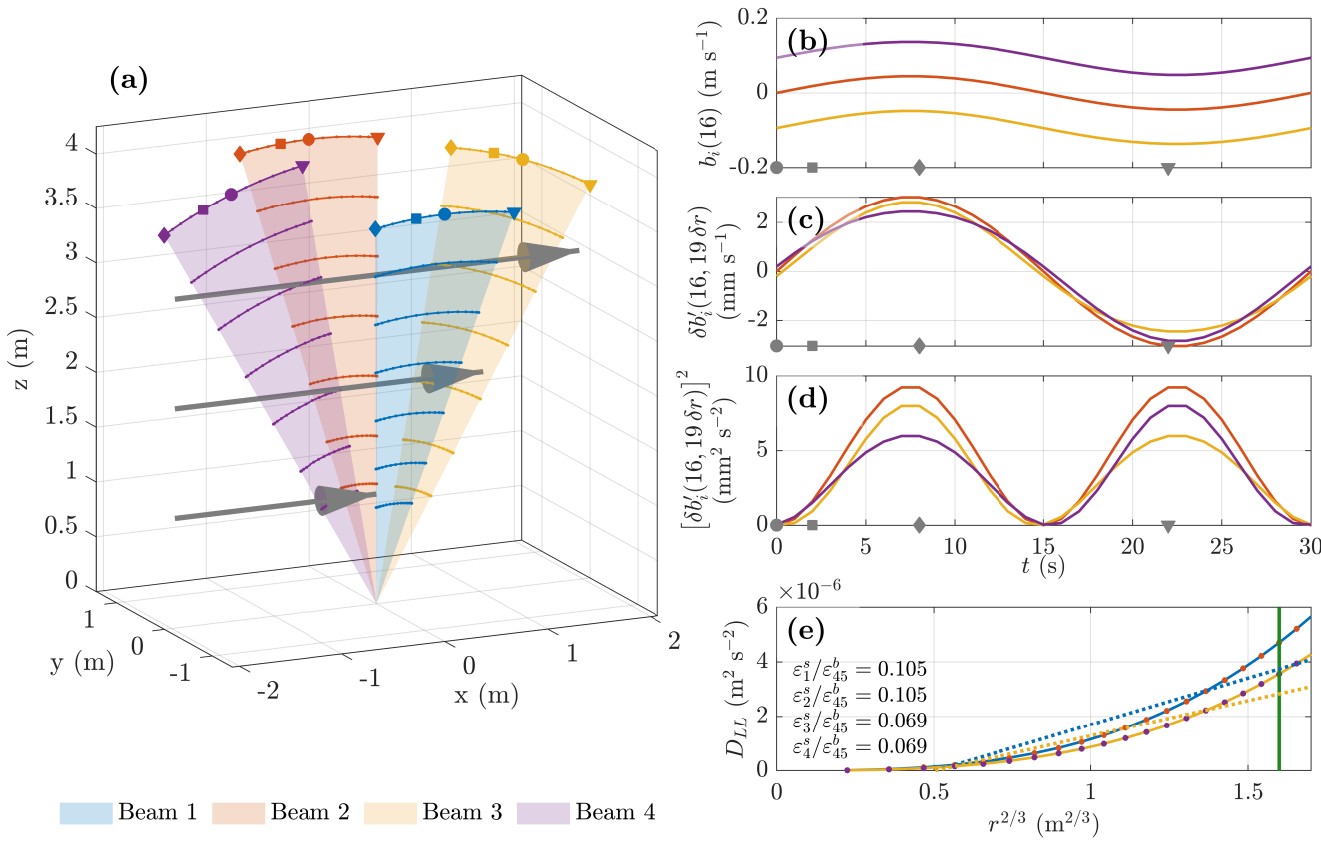

**Figure 9.** Illustration of pitch oscillation in a sheared current for $\phi_P(0) = 0°$, $\Delta\phi_P = 20°$ and $t_{\phi_P} = 30\,\mathrm{s}$, with $\phi_H(t) = \beta$ and $\phi_R(t) = 0°$ for all $t$. Panel details as per Figure 7.

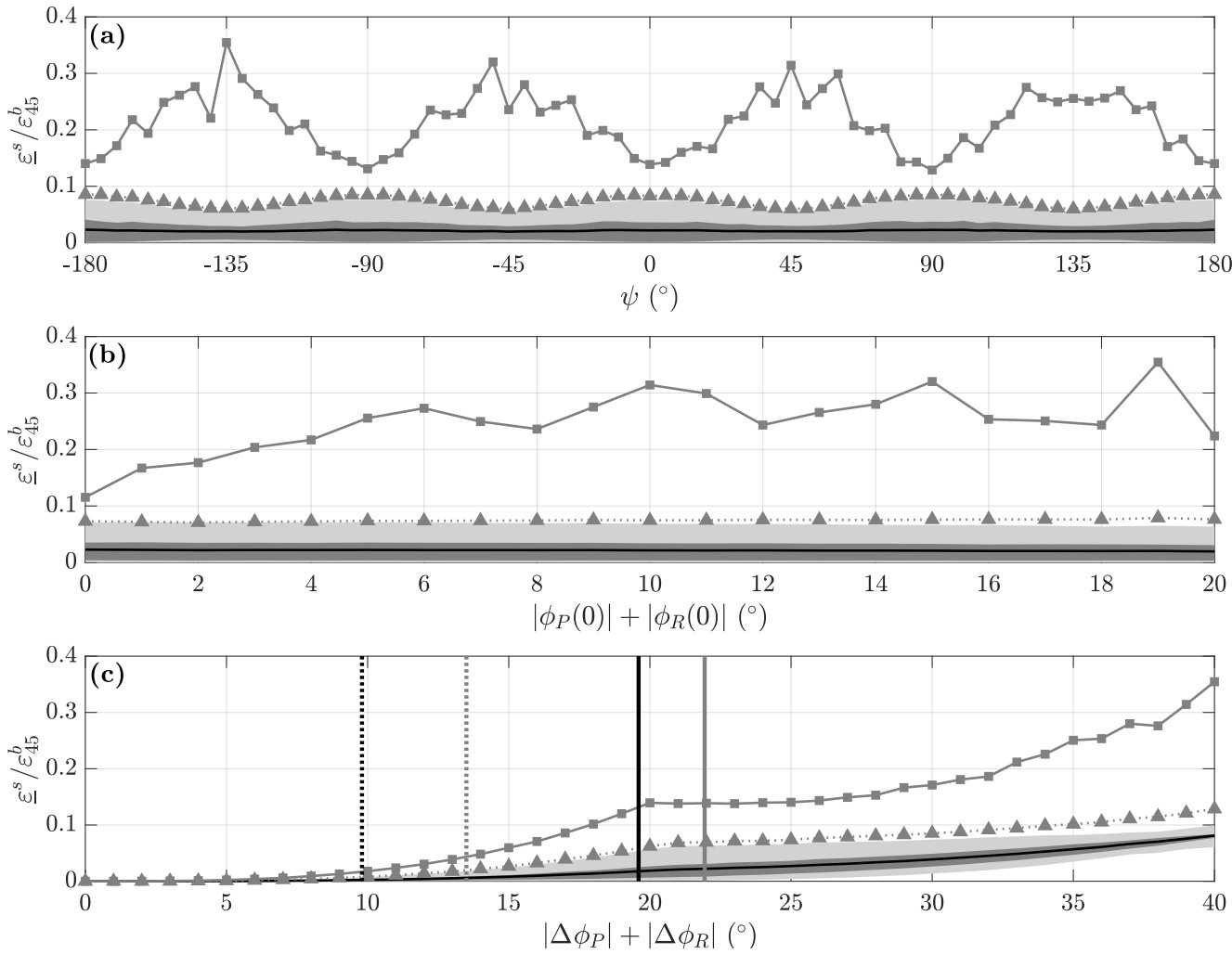

**Figure 10.** Normalised residual bias distribution for scenarios with tilt oscillation. Each panel shows the mean (black line), 25 % to 75 % range (dark grey shading) and 5 % to 95 % range (light grey shading) for the beam averaged normalised residual bias, $\underline{\varepsilon}^s/\varepsilon_{45}^b$, together with the 95th percentile (dotted line with triangle markers) and maximum (grey line with square markers) individual beam normalised residual bias, $\varepsilon_i^s/\varepsilon_{45}^b$ for scenarios aggregated by: panel (a) the heading difference angle, $\psi$; panel (b) the sum of the absolute initial pitch and roll, $|\phi_P(0)|+|\phi_R(0)|$; and panel (c) the sum of the absolute pitch and roll angular oscillation ranges, $|\Delta\phi_P|+|\Delta\phi_R|$. The vertical lines in panel (c) are the deployment 4 median (dotted line) and 90th percentile (solid line) $\Delta\phi_P + \Delta\phi_R$ values for the upper (grey) and middle (black) instruments from the observations described in section 3.

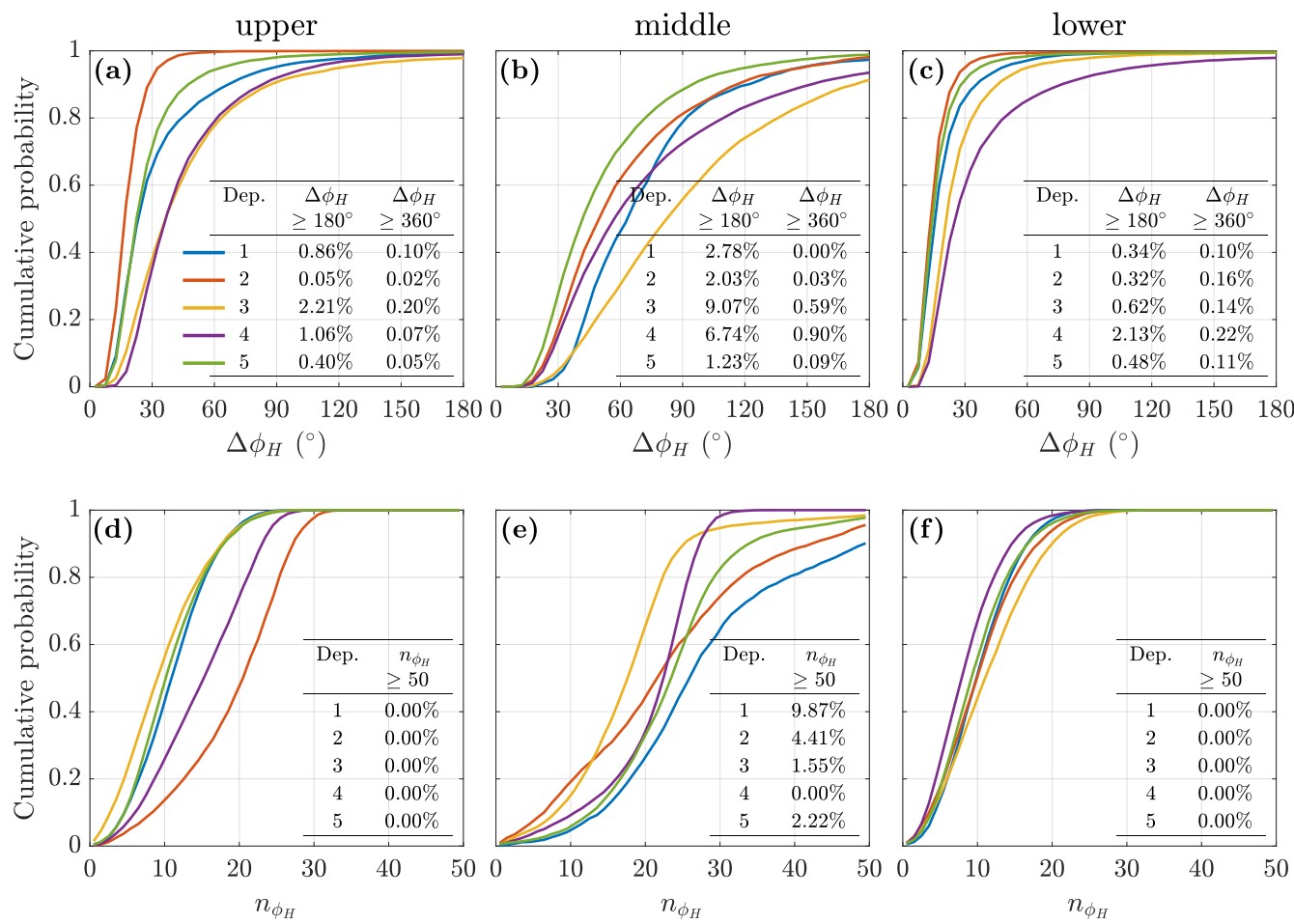

**Figure A1.** Cumulative probability of (a) - (c) burst heading range, $\Delta\phi_H$, and (d) - (f) mean number of oscillations per burst, $n_{\phi_H}$, by instrument and deployment, as per legend in panel (a).

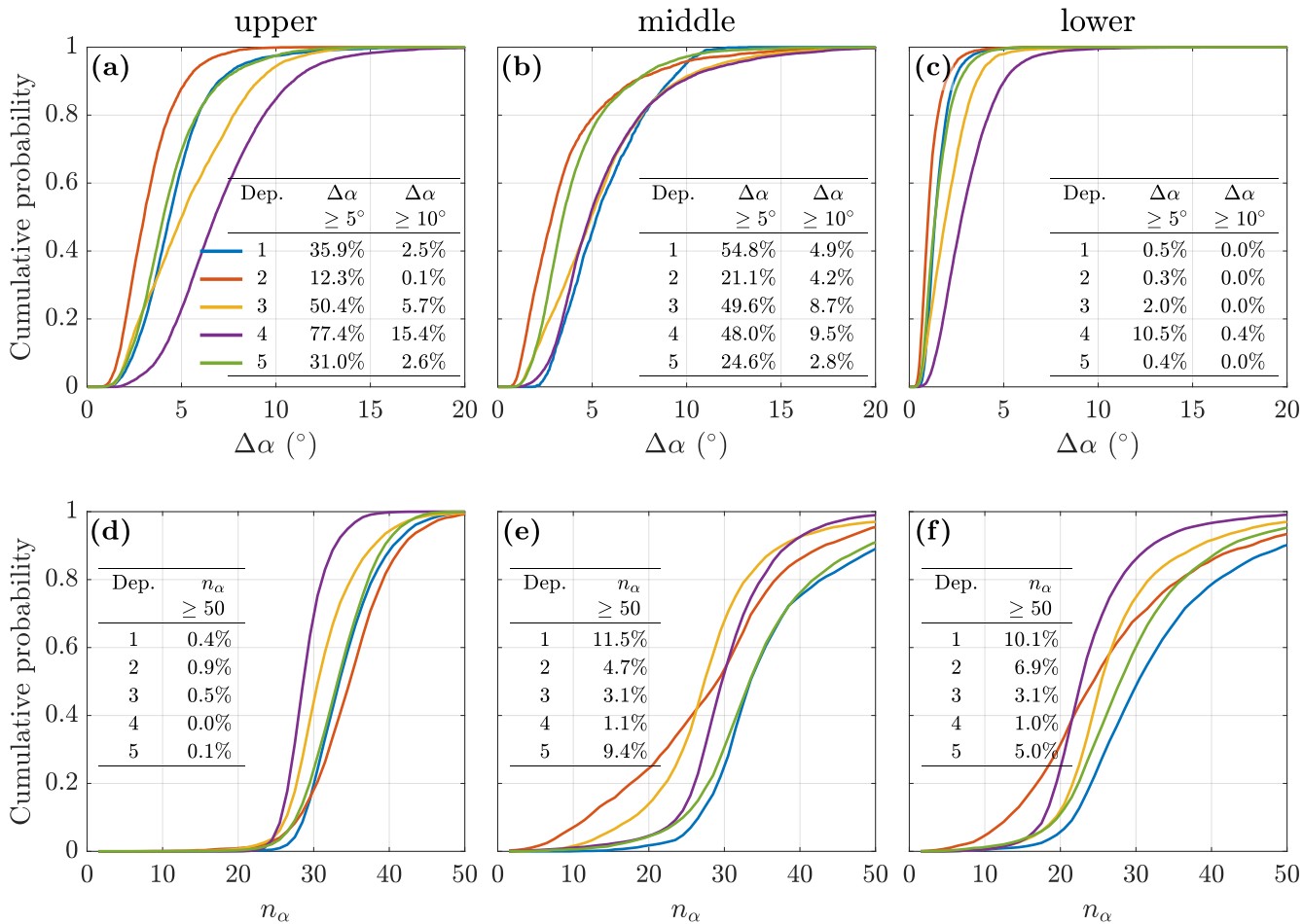

**Figure A2.** Cumulative probability of (a) - (c) burst tilt range, $\Delta\alpha$, and (d) - (f) mean number of tilt oscillations per burst, $n_\alpha$, by instrument and deployment, with line colour indicating the deployment as shown in panel (a); tables in panels (a) - (c) show the percentage of bursts when $\Delta\alpha$ exceeded 5° and 10°; tables in panels (d) - (f) show the percentage of bursts when $n_\alpha \geqslant 50$.

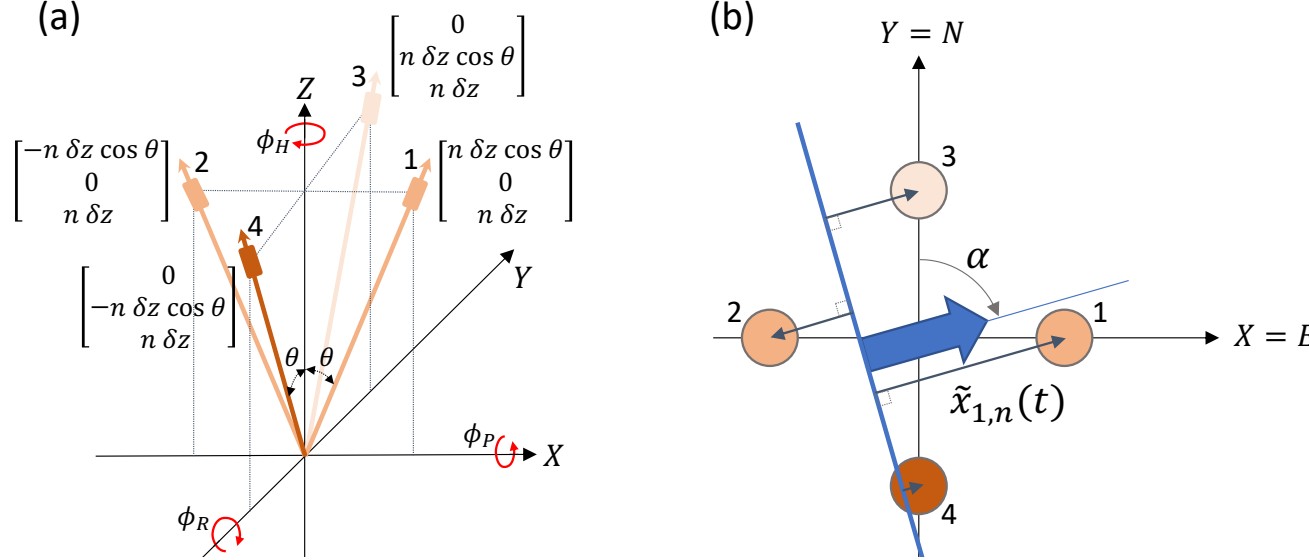

**Figure A3.** Geometry for synthesis of along-beam velocities. (a) $\mathbf{X}_i$ coordinate positions for bin $n$ in each beam with instrument upright ($\phi_P$ and $\phi_R$ both 0°) and heading angle $\phi_H = 0\,°\mathrm{N}$; (b) geometry for surface wave orbital velocity viewed from above with the block arrow showing the direction of wave propagation, $\alpha$ (°N); the blue line being a wave front; and the grey arrows indicating $\widetilde{x}_{i,n}(t)$, being the distance from the wave front in the direction of wave propagation of beam $i$ bin $n$ at time $t$.

| Dep. | Recovery Cruise | Deployment Date | Recovery Date | Bursts returned by instrument | | |
|------|-----------------|-----------------|---------------|--------|--------|--------|
| | | | | upper | middle | lower |
| 1 | JC105 | 27 March 2014 | 19 June 2014 | 5799 [a] | 1621 [b] | 5799 [a] |
| 2 | DY026 | 22 June 2014 | 21 August 2014 | 4332 | 3695 [c] | 4333 |
| 3 | DY018 | 22 August 2014 | 20 November 2014 | 6472 | 6472 | 6473 |
| 4 | DY029 | 22 November 2014 | 4 April 2015 | 9571 | 9572 | 9571 |
| 5 | DY033 | 11 April 2015 | 25 July 2015 | 7568 | 7567 | 7568 |

[a] no data post 15 June 2014 - memory full

[b] no data post 18 April 2014 - instrument stopped logging

[c] no data between 8 and 17 July 2014 - instrument stopped logging

**Table A1.** Turbulence mooring deployment periods.

| Dep. | Inst. | $n_{obs}$ | $\overline{z}$ | $\Delta\phi_H$ | | | | | $n_{\phi_H}$ | | | |
|---|---|---|---|---|---|---|---|---|---|---|---|---|
| | | | | $\overline{\Delta\phi_H}$ | 10% | 50% | 90% | $\geqslant 360°$ | $\overline{n_{\phi_H}}$ | $\leqslant 1$ | 50% | 90% |
| 1 | upper | 5799 | 19.2 | 36.2 | 15.2 | 25.8 | 68.6 | 0.10% | 11.1 | 1.10% | 11 | 18 |
| | middle | 1621 | 32.9 | 74.4 | 39.6 | 65.9 | 123.5 | 0.00% | 28.3 | 0.37% | 26 | 49 |
| | lower | 5799 | 49.4 | 22.9 | 10.9 | 17.9 | 37.6 | 0.10% | 10.2 | 1.21% | 10 | 17 |
| 2 | upper | 4332 | 19.6 | 21.1 | 12.5 | 19.2 | 30.7 | 0.02% | 19.1 | 0.46% | 20 | 27 |
| | middle | 3695 | 32.9 | 64.6 | 27.7 | 52.1 | 118.5 | 0.03% | 22.7 | 2.49% | 22 | 42 |
| | lower | 4333 | 50.0 | 18.7 | 10.5 | 15.8 | 26.4 | 0.16% | 10.4 | 2.86% | 10 | 18 |
| 3 | upper | 6472 | 19.7 | 50.6 | 19.7 | 38.9 | 89.7 | 0.20% | 9.7 | 4.64% | 9 | 18 |
| | middle | 6472 | 33.2 | 98.7 | 39.5 | 84.9 | 175.4 | 0.59% | 17.8 | 1.44% | 18 | 25 |
| | lower | 6473 | 50.4 | 30.0 | 14.7 | 23.4 | 48.1 | 0.14% | 11.6 | 2.53% | 11 | 20 |
| 4 | upper | 9571 | 22.3 | 48.9 | 23.0 | 39.1 | 85.7 | 0.07% | 14.9 | 0.72% | 15 | 23 |
| | middle | 9572 | 35.7 | 80.0 | 29.2 | 59.5 | 154.6 | 0.90% | 20.7 | 1.08% | 22 | 27 |
| | lower | 9571 | 50.8 | 41.6 | 15.8 | 27.6 | 79.3 | 0.22% | 8.5 | 2.60% | 8 | 15 |
| 5 | upper | 7568 | 19.5 | 30.8 | 15.6 | 24.9 | 49.8 | 0.05% | 10.8 | 1.36% | 10 | 18 |
| | middle | 7567 | 33.1 | 54.8 | 24.1 | 44.3 | 96.9 | 0.09% | 23.7 | 0.94% | 23 | 34 |
| | lower | 7568 | 50.3 | 20.8 | 10.8 | 16.5 | 30.8 | 0.11% | 9.8 | 1.88% | 9 | 17 |

**Table A2.** Heading statistics by deployment and instrument. Number of bursts, $n_{obs}$, and mean instrument depth, $\overline{z}$ m. For heading range, $\Delta\phi_H$: the mean, $\overline{\Delta\phi_H}$; 10th, 50th and 90th percentiles; and the percentage of bursts with $\Delta\phi_H \geqslant 360°$. For oscillation count per burst, $n_{\phi_H}$: the mean, $\overline{n_{\phi_H}}$; percentage of bursts with $n_{\phi_H} \leqslant 1$; and the 50th and 90th percentiles.

| Dep. | Inst. | $\overline{\lvert\langle\phi_P\rangle\rvert}$ | $\geqslant 5°$ | $\geqslant 10°$ | $\overline{\Delta\phi_P}$ | 10 % | 50 % | 90 % | $\overline{n_{\phi_P}}$ | 10 % | 50 % | 90 % |
|---|---|---|---|---|---|---|---|---|---|---|---|---|
|  |  |  | $\lvert\langle\phi_P\rangle\rvert$ |  | | $\Delta\phi_P$ | | | | $n_{\phi_P}$ | | |
| 1 | upper | 1.3 | 0.1 % | 0.00 % | 4.5 | 2.1 | 4.0 | 7.4 | 34.8 | 29 | 34 | 41 |
|  | middle | 2.8 | 7.6 % | 0.00 % | 6.8 | 3.3 | 6.1 | 11.6 | 35.5 | 26 | 33 | 50 |
|  | lower | 3.1 | 17.9 % | 0.78 % | 1.6 | 0.9 | 1.4 | 2.4 | 32.4 | 19 | 29 | 50 |
| 2 | upper | 3.7 | 16.8 % | 0.00 % | 3.4 | 1.6 | 3.0 | 5.6 | 34.5 | 27 | 34 | 42 |
|  | middle | 2.7 | 12.5 % | 1.62 % | 3.7 | 1.1 | 2.6 | 7.6 | 28.7 | 11 | 29 | 45 |
|  | lower | 3.4 | 18.0 % | 0.02 % | 1.2 | 0.6 | 1.0 | 2.0 | 29.8 | 15 | 27 | 48 |
| 3 | upper | 2.4 | 11.6 % | 0.83 % | 5.0 | 1.9 | 4.4 | 8.8 | 32.4 | 27 | 32 | 39 |
|  | middle | 1.9 | 8.7 % | 1.59 % | 5.5 | 2.0 | 4.8 | 9.8 | 28.7 | 17 | 28 | 39 |
|  | lower | 2.7 | 14.7 % | 1.79 % | 2.1 | 0.8 | 1.8 | 3.7 | 26.6 | 17 | 25 | 38 |
| 4 | upper | 2.6 | 6.9 % | 0.00 % | 7.6 | 3.7 | 7.0 | 12.0 | 29.0 | 25 | 29 | 33 |
|  | middle | 3.3 | 24.7 % | 2.84 % | 5.6 | 2.3 | 4.6 | 10.2 | 30.8 | 23 | 30 | 40 |
|  | lower | 3.6 | 25.3 % | 1.49 % | 3.0 | 1.4 | 2.6 | 5.0 | 24.0 | 16 | 23 | 33 |
| 5 | upper | 2.2 | 8.9 % | 0.01 % | 4.6 | 2.2 | 4.1 | 7.5 | 33.4 | 27 | 33 | 40 |
|  | middle | 2.6 | 14.3 % | 1.94 % | 4.0 | 1.7 | 3.2 | 7.2 | 35.6 | 24 | 34 | 51 |
|  | lower | 2.7 | 15.8 % | 1.41 % | 1.6 | 0.8 | 1.4 | 2.8 | 29.4 | 17 | 27 | 44 |

**Table A3.** Pitch statistics by deployment and instrument. For the absolute burst mean pitch, $\lvert\langle\phi_P\rangle\rvert$: deployment mean, $\overline{\lvert\langle\phi_P\rangle\rvert}$; and the percentage of bursts with $\lvert\langle\phi_P\rangle\rvert \geqslant 5°$ and $10°$. For the pitch burst range, $\Delta\phi_P$: the deployment mean, $\overline{\Delta\phi_P}$; and the 10[th], 50[th] and 90[th] percentiles. For the oscillations per burst, $n_{\phi_P}$: the deployment mean, $\overline{n_{\phi_P}}$; and the 10[th], 50[th] and 90[th] percentiles.

| Dep. | Inst. | $\overline{|\langle\phi_R\rangle|}$ | $|\langle\phi_R\rangle|$ $\geqslant 5°$ | $\geqslant 10°$ | $\overline{\Delta\phi_R}$ | $\Delta\phi_R$ 10% | 50% | 90% | $\overline{n_{\phi_R}}$ | $n_{\phi_R}$ 10% | 50% | 90% |
|---|---|---|---|---|---|---|---|---|---|---|---|---|
| | upper | 0.8 | 0.0% | 0.00% | 4.9 | 2.5 | 4.6 | 7.5 | 34.0 | 28 | 33 | 41 |
| 1 | middle | 4.6 | 38.3% | 3.95% | 4.6 | 2.5 | 4.3 | 7.2 | 36.8 | 27 | 34 | 52 |
| | lower | 1.5 | 2.6% | 0.00% | 1.6 | 0.8 | 1.3 | 2.5 | 34.9 | 23 | 31 | 53 |
| | upper | 1.8 | 4.6% | 0.07% | 3.2 | 1.6 | 2.9 | 5.2 | 35.8 | 28 | 36 | 44 |
| 2 | middle | 2.4 | 10.7% | 1.30% | 3.8 | 1.2 | 2.8 | 7.9 | 28.3 | 10 | 30 | 44 |
| | lower | 4.3 | 29.5% | 4.08% | 1.2 | 0.6 | 1.0 | 2.0 | 25.0 | 8 | 23 | 44 |
| | upper | 2.2 | 0.1% | 0.00% | 5.9 | 2.2 | 5.4 | 10.3 | 31.1 | 25 | 30 | 39 |
| 3 | middle | 2.5 | 10.9% | 1.38% | 5.6 | 1.8 | 4.8 | 10.2 | 28.2 | 17 | 28 | 39 |
| | lower | 1.6 | 4.2% | 0.00% | 2.2 | 0.9 | 1.9 | 3.8 | 29.0 | 20 | 26 | 41 |
| | upper | 3.5 | 19.7% | 0.01% | 6.8 | 3.5 | 6.4 | 10.8 | 29.6 | 25 | 29 | 34 |
| 4 | middle | 2.1 | 9.3% | 0.42% | 6.0 | 2.7 | 5.2 | 10.4 | 30.6 | 24 | 30 | 40 |
| | lower | 3.2 | 16.8% | 2.83% | 3.3 | 1.3 | 2.8 | 5.6 | 25.0 | 18 | 24 | 34 |
| | upper | 4.1 | 28.6% | 0.89% | 4.4 | 2.0 | 3.9 | 7.5 | 34.0 | 28 | 34 | 41 |
| 5 | middle | 3.6 | 22.6% | 3.86% | 4.3 | 1.7 | 3.5 | 7.8 | 35.4 | 24 | 34 | 50 |
| | lower | 4.0 | 29.1% | 1.02% | 1.5 | 0.7 | 1.3 | 2.7 | 30.8 | 20 | 29 | 44 |

**Table A4.** Roll statistics by deployment and instrument. For the absolute burst mean roll, $|\langle\phi_R\rangle|$: deployment mean, $\overline{|\langle\phi_R\rangle|}$; and the percentage of bursts with $|\langle\phi_R\rangle| \geqslant 5°$ and $10°$. For the roll burst range, $\Delta\phi_R$: the deployment mean, $\overline{\Delta\phi_R}$; and the 10[th], 50[th] and 90[th] percentiles. For the oscillations per burst, $n_{\phi_R}$: the deployment mean, $\overline{n_{\phi_R}}$; and the 10[th], 50[th] and 90[th] percentiles.