# Peer review of "Impact of ADCP motion on structure function estimates of turbulent kinetic energy dissipation rate"

_Ocean Science, 2021_

## Author Response (AR1)

**Response to Referees' Comments on Submission OS-2021-71**
**Impact of ADCP motion on structure function estimates of turbulent kinetic energy dissipation rate**

Brian Daniel Scannell        Yueng-Djern Lenn        Tom P. Rippeth

24th November 2021

Our responses to the referees' comments on our submission are as follows:

**Comments by Referee #1**

> This manuscript examines the use, and potential bias, of the structure function method applied to high frequency (1 Hz) data collected with a moored ADCP for over a year. It should be noted that the paper is limited to assessing the impact of ADCP motion on the estimates of the dissipation rate, and makes no attempt to describe or explain the seasonal variability in the dissipation rate itself. The paper is very well written, comprehensive, and timely given that improvements in ADCP technology have enhanced the ability to measure velocity fluctuations with these instruments.
>
> The manuscript presents two significant conclusions that are worthy of publication. First, the authors derive an equation for the bias introduced to the dissipation estimates due to the presence of background velocity shear. Second, the authors use synthetic data to demonstrate that the periodic motion of the ADCP adds only a small bias to the estimated dissipation rate (less than 10% for good mooring performance). Because the structure function method is based on velocity differences, the insensitivity to instrument motion is not surprising, but the manuscript helps to quantify the bias. These results will help to objectively quality control similar ADCP measurements. The authors also provide useful guidance on achieving better dissipation estimates from a moving ADCP.
>
> I recommend this manuscript be published with some minor improvements. The overall suggestions I have are:
>
> 1. There should be better connections drawn between the observational data and the synthetic data. There is some nice discussion starting on line 398, but I think more conclusions can be drawn, specifically in how these results would help in data processing and quality control of these specific measurements.
>
> - Can you quantify the approximate bias in the measurements from the results of the synthetic data and your measured shear and/or attitude variations? Are there measurements / intervals that you would reject in your turbulence analysis as a result?
>
> - What is this bias relative to the estimated dissipation rates? How does this bias compare to the minimum measurable dissipation rate achievable due to the Doppler noise?
>
> - Do your conclusions suggest that averaging the dissipation rates across all four beams minimizes the introduction of any bias?

We appreciate the referee's support for our manuscript and their helpful comments.

We will be seeking to publish a paper in the near future detailing the analysis of the data from the turbulence mooring referred to in this paper. This will address issues such as the level of bias introduced by both the motion of the ADCP and the orbital velocities forced by surface gravity waves when using the standard regression method, as well as the averaging of $\varepsilon$ estimates across bins and between beams.

We feel that including the additional analysis in the current paper would result in it becoming excessively long and would distract from the key issue we are trying to present, namely that when using the standard structure function methodology, the $\varepsilon$ estimates are potentially subject to bias due to the the motion of the ADCP in a sheared flow.

> 2. Include a review and a discussion about the results in comparison to other studies that used moving ADCPs. For example, surface drifters (SWIFT buoys and others) have recently been used to estimate the dissipation rate in tidal channels (Guerra et al. https://doi.org/10.1016/j.renene.2019.02.052 0960-1481/ and https://doi.org/10.1016/j.renene.2021.05.133).
>
> - These drifters commonly only use a central "vertical" beam to estimate the dissipation rate. Could your synthetic data analysis be extended to include the vertical beam? This would be an interesting application of your results.

We are grateful for the referee flagging up that this issue was not addressed in the original manuscript and for the references provided. We have addressed the basic points in the revisions to section 1, see lines 29-31, 45-46 and 63 of the revised manuscript.

We feel that extending the analysis in section 4 to include specific consideration of $\varepsilon$ estimates derived from the vertical beam data of surface-drifter mounted ADCP is beyond the scope of this paper. The synthesis of ADCP velocities we present assumes that the ADCP is essentially stationary (just changing orientation). Extending the analysis to allow for the variable motion of the ADCP moving with the current but influenced by wind and waves would require substantial development work and would need to be informed by appropriate observational experience and the associated data.

> 3. I think the paper would benefit from having some of the observational results moved from the appendix to the main body (e.g. Fig A1). This is a personal preference to help the reader get a feel for the data and its variability. This helps to motivate the paper and visualize key variables that are used in the analysis. (Specific comments below)

We have implemented this suggestion. The old Figure A1 is now Figure 4 and the associated text from Appendix B now forms section 3.2, starting at line 205.

> Specific Comments
>
> - Abstract: I suggest removing the sentence starting on line 3 about microstructure profilers. While these instruments are indeed the dominant method used to measure the dissipation rate, they are not used in this study so I do not think they need to be mentioned in the abstract. The introduction would be a better location to summarize the benefit of ADCPs over other technology (microstructure profilers, ADVs) for mooring deployments. Microstructure profilers require a background flow and consume a lot of power, whereas ADVs require significant correction for motion (compared to ADCPs).

We have implemented this suggestion. The relevant text has been removed from the Abstract and expanded text included in the Introduction starting at line 20.

> - Line 38: Can you quantify the smallest dissipation rate that can be measured with an ADCP?

This is an issue of considerable interest and while it isn't within the scope of the current article, we have included a brief comment in the introduction, starting at line 39. Quantifying the link between the uncertainty of the ADCP velocity measurements and the detection limit requires further work, which is complicated by the limited information published by manufacturers seeking to protect proprietary information.

- Line 90: "a function of the noise of the velocity of observations" is vague. Why not specify two times the Doppler noise variance like in Wiles et al (2006).

We have addressed this at lines 119 - 120.

- Line 92: "homogeneous" or "isotropic"?

Corrected to isotropic, see line 123.

- It would be useful to include a schematic of the mooring line in Section 3. This should show the relative positions and orientations of the ADCPs with respect to each other and the total water depth. It would also highlight that the middle instrument was in an open frame unlike the spherical buoys used for the top and bottom instruments. It is not critical to the results but helps visualize the configuration.

We have implemented this suggestion with the inclusion of the new Figure 3.

- Line 110: It would be useful to explicitly give the equation for $\varepsilon^b$ as a function of $S$ and $r_n$.

We are unable to provide a single equation for $\varepsilon^b$ as the gradient of the regression of $D_{LL}^b$ against $r^{\frac{2}{3}}$ depends on both $r_{\max}$ and $\delta r$. However, we have followed the referee's suggestion (below) of presenting example data graphically and we have sought to improve the clarity of the text, starting at line

- Line 114: "When using..." This sentence is confusing. The illustration and quantification in the paragraph that follows is helpful, but I think it would be beneficial to include a plot of the dependency of $\varepsilon^b$ on $S$, $r_n$ and $r_{\max}$. This is a key result of the manuscript that quantifies the bias as a function of shear and instrument configuration. Perhaps a figure of $\varepsilon^b$ versus $S$ for the three combinations of bin size and $r_{\max}$ (i.e. 0.1 m bins with $r_{\max}$=0.92, 0.1 m bins with $r_{\max}$=1.92, 0.2 m bins with $r_{\max}$=1.92) would be sufficient.

We have added new Figure 2 to illustrate the variation of $\varepsilon^b$. Since the dependence on $S$ is simply $\varepsilon^b \propto S^3$, we choose to show the variation of $\varepsilon^b$ and $r_{\max}$ for a range of $\delta r$ and two $S$ options. We believe that the amended text together with this new figure addresses the concerns of the referee.

- Section 3.1: I propose including Figure A.1 in this section to give a sample of the measured speeds and instrument attitude. This helps to visualize the variability within the 5-minute bursts and motivates the definition of the parameters used in sections 3.2 and 3.3. It may be useful to also include the dissipation rates in this sample time series (or elsewhere in the paper).

We have implemented this suggestion, with the old Figure A1 and associated text from Appendix B now forming Figure 4 and section 3.2, starting at line 205. As previously noted, we feel that presenting dissipation data is beyond the scope for the current work. The objective here was simply to illustrate the motion of ADCP for instruments at different depths and in different mountings under a wide range of environmental conditions.

- Figures 1 and 2: I find these plots a little hard to interpret, but the text does a good job of summarizing the major conclusions. I wonder if simpler plots of the coloured parameters (e.g. oscillation range, oscillations per burst) versus $U$ and $H_{m_0}$ would be more intuitive and better show the trends.

We explored alternative presentations of this data but concluded that we prefer to stick with the existing format as alternatives quickly ended up requiring multiple plots.

- Section 4: Figures 3-6 are well described, but some additional references to line types and colours would help guide the reader. My suggestions are in italics

  - Line 295: "to increase with $\Delta\phi_H$", *as indicated by the varying line types*
  - Line 310: "beam 3" *(yellow)*
  - Line 313: "for all bins in both beams" *(red line overlying blue line)*
  - Line 353: "range of the beam average values" *(light gray shading)*
  - Line 365: "maximum of 0.09" *(black line)*

Each of these suggestions has been implemented, see lines 371, 392, 394, 438 and 449 respectively.

- Figures 4 and 6: Is it possible to show the range of values from the observations in these plots of the synthetic data? (e.g. the 50 or 90% $\Delta\phi_H$ values presented in table A2 could be plotted as vertical bars in Fig 4b and Fig 4c for each of the ADCPs). This might help show that there is expected to be only small biases in the estimates of the dissipation rate from the measurements.

This suggestion has been implemented in the revised Figures 8 and 10.

- Line 382: "separation distance" (eq. 8) [i.e. add referemce to equation]

This suggestion has been implemented, see line 487.

- Line 413: can you clarify what you mean by "the linear length scale dependence of the velocity difference between bins"? Are you suggesting that the method used to eliminate waves (eq 6) also handles shear?

We consider the fact that the equation (6) regression proved wholly effective in removing the potential bias in the synthesised data analysis is an important conclusion of this work. We apologise that we failed to communicate this clearly in the earlier version of the manuscript. We have included new section 4.5, starting at line 458, and have made small changes in the Discussion, starting from line 512, to better communicate this point.

Technical Corrections

This paper is extremely well written with very few technical errors. The couple I noticed include:

- line 53: "us" should be "is"

- line 84: "anticipate" should be "anticipates"

We have made the first change, see line 68. For the second point, since "hypotheses" is plural, we consider "anticipate" to be correct, see line 109.

**Comments by Referee #2**

> This is an interesting and very clearly written paper. It reaches well supported conclusions of value to the many ADCP observations of the ocean. The conclusions are logical and produce a concrete list of recommendations based on their evidence. I enjoyed the combination of observational evidence and synthetic data, which produce a useful combined message. I also found the geometric problem of how to produce the synthetic data interesting in itself and Appendix C lays this out in a good level of detail for reproduction. There are a few minor details that the authors should consider. However, the paper is otherwise ready for publication as is.
>
> Minor Comments
>
> line 118-121 : Some context for the numbers quoted here would be very helpful to those less familiar with ADCP measurements than the authors. Are the $20°$ beam angle, bin size and number of bins typical? Is the shear value large or moderate? How do the values of epsilon compare to the expected values?

We have addressed this point with the modified text in section 2, see lines 145 to 151, to accompany the new Figure 2.

> line 122-124 : Can an error estimate be provided a priori? Or is the error unconstrained at this point in the paper?

Other than determining the maximum potential bias i.e. $\varepsilon^b$, no estimate is possible. Previously the potential for bias had not been recognised. The objective of this analysis was to seek to quantify the potential bias and to identify the key factors influencing it.

> line 145 : When I got to this point I realised that I wasn't quite as sure on the geometry of the problem as I thought. There are quite a few angles involved, due to the inclination of the ADCP beams, the heading angle, etc. I think it would be a good idea to lay this geometry out clearly in some form of schematic. It would help to visualise the rotations of the ADCP if I could refer back to such a figure later in the paper.

We have added the new Figure 1 to clarify the ADCP geometry and rotation axes.

> line 263 & Figures 3 & 5 : There's lots of information in these figures and they are very well drawn. It would be helpful to have something in the figures to show the point in the oscillation cycle that the ADCP is at in panels b-d. This would tie them to the markers in panel a. I think this would help with understanding some interesting details of the figures, such as why the oscillation frequency might differ between beams. Perhaps reduce the number of cycles shown in panels b-d and include the circle/square/diamond/triangle systems for the first oscillation?

We have implemented this suggestion, so the revised Figures 7 and 9 only show the first 30 s of the burst and we have included the bin 30 time markers from panel (a) on the time axis in panels (b), (c) and (d) in each figure. We have also made associated small changes to the text.

> line 284-286 & 340-346 : The ranges used in the different scenarios are very clearly laid out. However, there isn't any information on why they were chosen. Are these values pulled from the observations? Or from reasonable values based on experience?

We have addressed this in lines 360 to 362 and 428 to 430 respectively.

Typos

I noticed an extraordinarily low number of typos.

line 53 : us $->$ is

Corrected, see line 68.

line 382: , is $->$ is

Corrected, see line 486.

---

## Author Response (AR2)

**Response to Referees' Comments on Submission OS-2021-71 - revision 2**
**Impact of ADCP motion on structure function estimates of turbulent kinetic energy dissipation rate**

Brian Daniel Scannell        Yueng-Djern Lenn        Tom P. Rippeth

15th December 2021

Our responses to the referees' comments on our submission are as follows:

**Comments by Referee #1**

1) In the opening paragraph of the introduction, the reference to Lueck 2016 is a non-peer reviewed report. Since the sentence is referring to microstructure profilers, a better reference may be the following review paper :

- Lueck, R. G., Wolk, F., & Yamazaki, H. (2002). Oceanic velocity microstructure measurements in the 20th century. Journal of Oceanography, 58(1), 153–174. https://doi.org/10.1023/A:1015837020019

I also think the authors would be amiss to neglect to mention the use of gliders for microstructure measurements. They do not require a surface vessel, but are still limited in duration in comparison to ADCP measurements (days to weeks). Some possible references include are:

- Fer, I., Peterson, A. K., & Ullgren, J. E. (2014). Microstructure measurements from an underwater glider in the turbulent Faroe Bank Channel overflow. Journal of Atmospheric and Oceanic Technology, 31(5), 1128–1150. https://doi.org/10.1175/JTECH-D-13-00221.1

- Scheifele, B., Waterman, S., Merckelbach, L., & Carpenter, J. R. (2018). Measuring the Dissipation Rate of Turbulent Kinetic Energy in Strongly Stratified, Low-Energy Environments: A Case Study From the Arctic Ocean. Journal of Geophysical Research: Oceans, 123(8), 5459–5480. https://doi.org/10.1029/2017JC013731

- Schultze, L. K. P., Merckelbach, L. M., & Carpenter, J. R. (2017). Turbulence and Mixing in a Shallow Shelf Sea From Underwater Gliders. Journal of Geophysical Research: Oceans, 122(11), 9092–9109. https://doi.org/10.1002/2017JC012872

We have revised the opening paragraphs of section 1 to address both the specific points raised and to recognise other relevant work; see lines 20 – 39.

2) Figure 2: I really like the addition of this figure. If I'm interpreting it correctly, it suggests that the bias of order $1 \times 10^{-8}\,\mathrm{W\,kg^{-1}}$ is comparable to the dissipation rate itself. I think this deserves a comment after line 151.

We are glad that the figure is helpful and have included additional text as suggested; see lines 158 – 159.

3) Typos:

- Line 53: "beams" should be "beam"

Corrected; see line 60.

- Line 147: you say "between 5 and 25 bins", but this isn't true in Figure 2 for all $\delta r$ (e.g. $\delta r > 0.2$). It might be easier to say "$r_{\max}$ varied between $0.5\,\mathrm{m}$ and $5\,\mathrm{m}$"

The different markers in Figure 2 actually relate to varying $\delta z$ rather than $\delta r$ (although the two are directly related as $\delta r = \delta z / \cos\theta$), which may explain the reviewer's comments. However, we have amended the text as suggested to clarify the issue; see line 154.

- Line 382: "fromt he" should be "from the"

A frustratingly recurrent typo that I apologise for not spotting. Corrected; see line 392.

**Comments by Referee #2**

Typos

- line 48 : double "the"

Corrected; see line 55.

- line 54 : "by any shear" $->$ "of any shear"?

Corrected; see line 60.

- line 361: "section'3" $->$ "section 3"

Corrected; see line 371.

- line 382 : "fromt he" $->$ "from the"

Corrected; see line 392.